# Dynamics Generalisation in Reinforcement Learning via Adaptive Context-Aware Policies

**Michael Beukman**[1,2*]    **Devon Jarvis**[1,3]
**Richard Klein**[1]    **Steven James**[1]    **Benjamin Rosman**[1]

[1]University of the Witwatersrand    [2]University of Oxford    [3]University College London

## Abstract

While reinforcement learning has achieved remarkable successes in several domains, its real-world application is limited due to many methods failing to generalise to unfamiliar conditions. In this work, we consider the problem of generalising to new transition dynamics, corresponding to cases in which the environment's response to the agent's actions differs. For example, the gravitational force exerted on a robot depends on its mass and changes the robot's mobility. Consequently, in such cases, it is necessary to condition an agent's actions on extrinsic state information **and** pertinent contextual information reflecting how the environment responds. While the need for context-sensitive policies has been established, the manner in which context is incorporated architecturally has received less attention. Thus, in this work, we present an investigation into how context information should be incorporated into behaviour learning to improve generalisation. To this end, we introduce a neural network architecture, the *Decision Adapter*, which generates the weights of an adapter module and conditions the behaviour of an agent on the context information. We show that the Decision Adapter is a useful generalisation of a previously proposed architecture and empirically demonstrate that it results in superior generalisation performance compared to previous approaches in several environments. Beyond this, the Decision Adapter is more robust to irrelevant distractor variables than several alternative methods.

## 1   Introduction

Reinforcement learning (RL) is a powerful tool, and has displayed recent success in several settings [1–4]. However, despite its potential, RL faces a significant challenge: generalisation. Current agents and algorithms struggle to perform well beyond the narrow settings in which they were trained [5–7], which is a major limitation that hinders the practical application of RL. For RL algorithms to be effective in real-world scenarios, they must be capable of adapting to changes in the environment and performing well in settings that are different, but related, to those they trained on [7].

To illustrate this point, consider a legged robot trained to walk on a tiled floor. While the agent would perform well in these conditions, it will likely struggle in other reasonable settings, such as walking on asphalt or when carrying cargo. When walking on another surface, the friction characteristics would differ from what the robot trained on, which may cause it to slip. When carrying additional weight, the robot is heavier, with a potentially different center of mass. This may mean that the robot must exert more force to lift its legs or be more careful to prevent falling over. This limitation is far from ideal; we want our agents to excel in tasks with minor differences in dynamics (the effect of the agent's actions) without requiring additional training.

---

*Correspondence to `mbeukman@robots.ox.ac.uk`. Work done while at the University of the Witwatersrand, now at the University of Oxford.

37th Conference on Neural Information Processing Systems (NeurIPS 2023).

Several methods have been proposed to address this problem, including training a single agent in various settings in the hope that it learns a generalisable and robust behaviour [3, 8–11]. However, this approach is not without its drawbacks, as it fails when variations are too large for a single behaviour to perform well in all settings [12, 13]. For example, using the same behaviour when walking unencumbered and under load may lead to suboptimal performance in one or both cases.

This observation has led to other methods that instead use contextual information to choose actions, allowing the agent to adapt its behaviour to the setting it is in. For instance, if the agent knows when it is walking unencumbered vs. carrying something heavy, it can perform slightly differently in each case [14]. The *context* differs conceptually from the *state* information, which generally includes data that the robot observes using its sensors, such as its joint angles, acceleration, and camera views. The crucial distinction between context and state is that context changes at a longer timescale compared to state; for instance, the floor friction and additional mass remain the same for a long time, whereas the robot's joint angles change after every action [15]. Currently, the predominant approach is to use context as part of the state information, ignoring the conceptual difference between these two aspects [16–18]. However, this may lead to the agent confounding the context and state, and thus exhibit worse generalisation [15]. This problem can be exacerbated in real-world settings, where identifying which variables affect the dynamics may be challenging [18]. This may lead to several irrelevant variables being added to the context, further expanding the state space and making learning more difficult.

To address this problem, we introduce an approach to incorporating context into RL that leads to improved generalisation performance. Our method separates the context and state, allowing the agent to decide how to process the state information based on the context, thus adapting its behaviour to the setting it is in. Our experimental results demonstrate the effectiveness of our approach in improving generalisation compared to other methods. We show that our approach outperforms (1) not incorporating context information at all; (2) simply concatenating context and state; and (3) competitive baselines. We also demonstrate that our approach is more robust to irrelevant distractor variables than the concatenation-based approach across multiple domains. We further provide a theoretical characterisation of problems where incorporating context is necessary and empirically demonstrate that a context-unaware method performs poorly.[1]

## 2   Background

Reinforcement learning problems are frequently modelled using a Markov Decision Process (MDP) [19, 20]. An MDP is defined by a tuple $\langle \mathcal{S}, \mathcal{A}, \mathcal{T}, \mathcal{R}, \gamma \rangle$, where $\mathcal{S}$ is the set of states, $\mathcal{A}$ is the set of actions. $\mathcal{T} : \mathcal{S} \times \mathcal{A} \times \mathcal{S} \to [0, 1]$ is the *transition function*, where $\mathcal{T}(s'|s, a)$ specifies the probability of ending up in a certain state $s'$ after starting in another state $s$ and performing a specific action $a$. $\mathcal{R} : \mathcal{S} \times \mathcal{A} \times \mathcal{S} \to \mathbb{R}$ is the *reward function*, where $R(s_t, a_t, s_{t+1}) = R_{t+1}$ specifies the reward obtained from executing action $a_t$ in a state $s_t$ and $\gamma \in [0, 1]$ is the environment discount factor, specifying how short-term and long-term rewards should be weighted. The goal in reinforcement learning is to find a policy $\pi : \mathcal{S} \to \mathcal{A}$ that maximises the *return* $G_t = \sum_{k=0}^{\infty} \gamma^k R_{t+k+1}$ [21].

We consider the *Contextual Markov Decision Process* (CMDP) [15] formalism, which is defined by a tuple $\langle \mathcal{C}, \mathcal{S}, \mathcal{A}, \mathcal{M}', \gamma \rangle$, where $\mathcal{C}$ is the *context space*, $\mathcal{S}$ and $\mathcal{A}$ are the state and action spaces respectively, and $\mathcal{M}'$ is a function that maps a context $c \in \mathcal{C}$ to an MDP $\mathcal{M} = \langle \mathcal{S}, \mathcal{A}, \mathcal{T}^c, \mathcal{R}^c, \gamma \rangle$. A CMDP thus defines a *family* of MDPs, that all share an action and state space, but the transition ($\mathcal{T}^c$) and reward ($\mathcal{R}^c$) functions differ depending on the context. Our goal is to train on a particular set of contexts, such that we can generalise well to another set of evaluation contexts. We focus solely on generalising over changing dynamics and therefore fix the reward function, i.e., $\mathcal{R}^c = \mathcal{R}, \forall c \in \mathcal{C}$.

## 3   Related Work

Generalisation in RL is a widely studied problem, with one broad class of approaches focusing on *robustness*. Here, a single policy is trained to be robust to changes that may occur during testing, i.e., to perform well without failing catastrophically [3, 6, 8, 22–29]. Many of these approaches are successful in this goal and can generalise well when faced with small perturbations. However, using one policy for multiple environments (or contexts) may lead to this policy being conservative, as it cannot exploit the specifics of the current environment [12]. Furthermore, these policies are limited

---

[1]We publicly release code at https://github.com/Michael-Beukman/DecisionAdapter.

to performing the same action when faced with the same state, regardless of the setting. This may be suboptimal, especially when the gap between the different settings widens [13, 30].

This observation motivates the next class of methods, context-adaptive approaches. These techniques often use a version of the CMDP [15] formalism, and use the context to inform the agent's choice of action. This allows agents to exhibit different behaviours in different settings, which may be necessary to achieve optimality or generalisation when faced with large variations [13]. There are several ways to obtain this context, such as assuming the ground truth context [13], using supervised learning to approximate it during evaluation [14, 31], or learning some unsupervised representation of the problem and inferring a latent context from a sequence of environment observations [18, 32–34].

Despite the progress in inferring context, how best to incorporate context has received less attention. The most prevalent approach is to simply concatenate the context to the state, and use this directly as input to the policy [16–18, 35–38], which is trained using methods such as Soft Actor-Critic [39] or Proximal Policy Optimisation [40]. This ignores the conceptual difference between the state and context, and may lead to worse generalisation [15]. This approach may also be limited when there are many more context features compared to state features [41]. While concatenation is the most common, some other approaches have also been proposed. For instance, Biedenkapp et al. [41] learn separate representations for context and state, and then concatenate the representations near the end of the network. They also use another baseline that concatenates a static embedding of the context features to the state. Both of these techniques generalise better than the concatenation approach. Another method, FLAP [42], aims to improve generalisation by learning a shared representation across tasks, which is then processed by a task-specific adapter in the form of a linear layer. Given state $s$, the policy network outputs features $\phi(s) \in \mathbb{R}^d$, which are shared across tasks. The final action is chosen according to $W_i \phi(s) + b_i$, with a unique weight matrix $W_i$ and bias vector $b_i$ which are learned separately for each task. Concurrently, a supervised learning model is trained to map between transition tuples $(s_t, a_t, r_{t+1}, s_{t+1})$ and these learned weights. At test time, this supervised model generates the weights, while the policy network remains fixed. FLAP generalises better to out-of-distribution tasks compared to other meta-learning approaches [36, 43, 44]. However, this approach requires a separate head for each task, which may scale poorly if we have many tasks or see each task only a few times.

One promising approach that has been used in other fields is Feature-wise Linear Modulation (FiLM) [45, 46]. In this method, features in one modality (e.g., natural language text) linearly modulate the neural-network features obtained from another modality (e.g., visual images). An example of this would be processing a visual scene, modulated by different natural language questions which would lead to a different final answer. In RL, Benjamins et al. [13] introduce `cGate`,[2] which follows a similar procedure to FiLM; in particular, the neural network policy receives the state as input and outputs an action. Before the final layer, however, the context is first transformed by a separate network, and then used to modulate the state-based features. This approach showed promise, but restricted the context features' effect on the state-based features to be linear.

Another research area that has garnered more attention recently is learning general foundation models for control [48–50]. Inspired by recent work in fields such as natural language processing [51] and computer vision [52], these models aim to provide a general foundation that can be easily fine-tuned for particular tasks of interest. For instance, Gupta et al. [53] learns a general controller for several different robot morphologies. The robot's configuration is encoded and passed to a transformer [54], allowing one model to control a wide variety of morphologies. While these works directly train policies, Schubert et al. [55] instead train a dynamics model, and use a standard planning technique— model-predictive control [56–58]—to choose actions. They find that this approach achieves better zero-shot generalisation compared to directly learning a policy. On a different note, Sun et al. [59] pre-train a foundation model that predicts observations and actions. Then, during fine-tuning, they train a policy to perform a particular task using the pre-trained representations, which enables the use of either imitation learning or reinforcement learning in the downstream fine-tuning stage.

Finally, much work in *meta-reinforcement learning* [60] relates to the problem of generalisation. Meta-learning approaches generally aim to use multiple tasks during training to *learn how to learn*; this knowledge can then be used during testing to rapidly adapt to a new task [43, 60–67]. In this problem setting, Beck et al. [68] use a recurrent model to encode the current task, and generate the weights of the agent policy based on the task encoding. Sarafian et al. [69] take a similar approach, but

---

[2]Benjamins et al. [13] introduced the `cGate` method, and then published a revised version of the paper—not containing `cGate`—under the same name. We therefore cite both versions [13, 47].

instead generate the weights using the environment state, and processing the encoded context using the generated network. Another approach, MAML [44] aims to learn the weights of a neural network such that, for any new task, performing only a few gradient updates would lead to a high-performing task-specific network. While meta-learning can lead to highly adaptable agents, many methods require multiple episodes of experience in the target domain [36, 44], and may therefore be less well-suited to the zero-shot setting we consider [7].

## 4 Theoretical Intuitions

Now, while only a context-conditioned policy is guaranteed to be optimal on a general CMDP [47], in many cases simply training an unaware policy on a large number of diverse environments can lead to impressive empirical generalisation [5]. In this section, we characterise and unify these two observations. In particular, for a specific problem setting—where the context defines the target location an agent must travel to—we find that:

(i) For some context sets, an unaware policy will perform arbitrarily poorly on average, due to it being forced to take the same action in the same state, regardless of context.

(ii) However, for other context sets—where the different contexts are similar enough—an unaware policy can perform well on average, as it can simultaneously solve each task.

We defer the formal statement of this theorem and its proof to Appendix A, but here we briefly provide some intuition. Overall, our results rely on the fact that a context-unaware agent *must* perform the same action in the same state, regardless of context. Therefore, in a setting where we have different goals (indicated by the context), and making progress on one goal leads to making negative progress on another, a context-unaware agent cannot simultaneously perform well on both of these contexts. By contrast, if the goals are *aligned* in the sense that making progress on one leads to progress on another, a context-unaware agent can perform well on average. A context-aware agent, however, can always choose the correct action in either scenario.

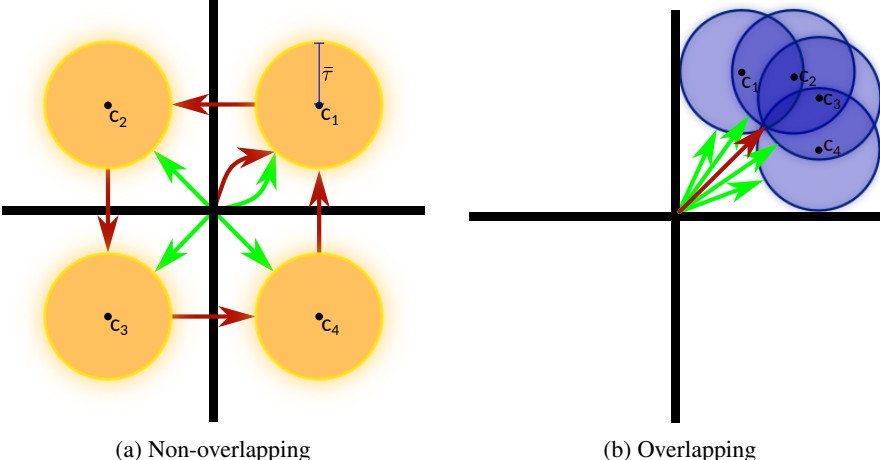

(a) Non-overlapping      (b) Overlapping

Figure 1: An illustration of the two cases discussed above, with the agent's start state being the origin. The agent will receive a reward if it enters the circle corresponding to the current context and zero reward otherwise. In (a) there is no overlap between the goals, so an unaware policy (red) must visit each goal in sequence, whereas a context-aware policy (green) can go directly to the correct goal. In (b), the goals overlap, so an unaware policy can go directly to the joint intersection point of the goals, leading to only a slightly suboptimal value function compared to the optimal context-aware policy.

Fig. 1 visually illustrates these two cases and summarises the implications. In Fig. 1a, if there is no overlap between goals, an unaware policy is forced to visit each one in sequence. This is because it must perform the same action in the same state, regardless of the context, meaning that it cannot always travel to the correct goal. This policy will therefore have a low value function compared to the optimal context-aware policy. By contrast, in Fig. 1b, a single unaware policy can directly go to the intersection of all the goals, leading to only slightly less return than the optimal context-aware

policy. While the formal result in Appendix A merely shows the existence of some problems with these properties, we believe the concept generalises to more complex settings. In particular, if the problem exhibits some structure where the same policy can simultaneously make progress on each context, an unaware model can perform well. However, if making progress on one goal leads to the agent moving away from another, a context-unaware policy can at best perform arbitrarily worse than a context-aware one, and incorporating context is crucial. We empirically substantiate these findings in Section 7 and formally consider this example in Appendix A. Finally, see Appendix B for more details about how these theoretical results connect back to our empirical observations.

# 5 The Decision Adapter

An adapter is a (usually small) neural network with the same input and output dimensions. This network can be inserted between the layers of an existing primary network to change its behaviour in some way. These modules have seen much use in natural language processing (NLP) [70–74] and computer vision, usually for parameter-efficient adaptation [75–81]. In these settings, one adapter is usually trained per task, and the base model remains constant across tasks. Our method is inspired by the use of adapters in these fields, with a few core differences.

First, since our aim is zero-shot generalisation, having a separate adapter module for each task (corresponding to context in our case) provides no clear means for generalisation. Relatedly, since tasks are generally discrete in NLP, but we allow for continuous contexts, this approach would be problematic. Furthermore, the number of learnable parameters increases with the number of training contexts, presenting a computational limitation to the scalability. To circumvent these problems, we use a *hypernetwork* [82] to generate the weights of the adapter module based on the context. This allows the agent to modify its behaviour for each context by having a shared feature-extractor network which is modulated by the context-aware adapter. Moreover, since the hypernetwork is shared, experience gained in one context could transfer to another [72]. Second, instead of having separate pre-training and adaptation phases as is common in NLP [76], we do not alter the standard RL training procedure at all. We change only the architecture and train normally—allowing for the easy integration of our approach into standard RL libraries. Finally, our method is agnostic to the exact RL algorithm and can be applied to most off-the-shelf algorithms without much additional effort.

**Algorithm 1** Decision Adapter—Changes to the standard forward pass in blue.

1: **procedure** ADAPTERFORWARD($s \in \mathcal{S}, c \in \mathcal{C}$)
2:     $x_1 = s$
3:     **for** $i \in \{1, 2, \ldots, n\}$ **do**
4:         **if** $A_i \neq null$ **then**
5:             $\theta_A^i = H_i(c)$     // Generate Weights
6:             $x_i' = A_i(x_i|\theta_A^i)$   // Forward Pass
7:             $x_i = x_i + x_i'$     // Skip connection
8:         **end if**
9:         $x_{i+1} = L_i(x_i)$
10:     **end for**
11:     **return** $x_{n+1}$
12: **end procedure**

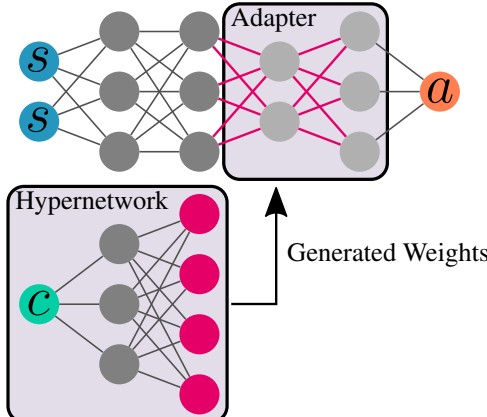

Figure 2: An illustration of our network architecture, the Decision Adapter, alongside pseudocode for the forward pass. The bottom network is the hypernetwork, which generates the magenta weights for the top, primary, network. The light grey nodes in the top network are added by the adapter.

To formally describe our approach (illustrated in Fig. 2), suppose we have a state $s \in \mathcal{S}$, context $c \in \mathcal{C}$, and a standard $n$-layer fully-connected neural network architecture that predicts our action $a$. Here $x_{i+1} = L_i(x_i) = \sigma_i(W_i x_i + b_i)$, $x_1 = s$ and $x_{n+1} = a$. We add an adapter module $A_i$ between layers $L_{i-1}$ and $L_i$. This adapter module is a multilayer neural network $A_i : \mathbb{R}^{d_i} \to \mathbb{R}^{d_i}$ consisting of parameters $\theta_A^i \in \mathbb{R}^{P_i}$.[3] These parameters are generated by the corresponding context-conditioned

---

[3]$d_i$ is the output dimension of layer $L_{i-1}$ and $P_i$ is the number of parameters in the adapter.

hypernetwork $H_i : \mathcal{C} \to \mathbb{R}^{P_i}$ (line 5 in Algorithm 1) and reshaped to form multiple weight matrices. Then, before we process $x_i$ using layer $L_i$, we pass $x_i$ through the adapter module $A_i$ (line 6) defined by the generated weights and biases to obtain $x_i' = A_i(x_i|\theta_A^i)$. Furthermore, akin to the application of adapters in other fields [75, 76], we add a skip connection, by setting $x_i = x_i + x_i'$ (line 7). This updated $x_i$ is then the input to layer $L_i$ (line 9). Additionally, we could have multiple adapter modules in the primary network, with at most one module between any two consecutive layers (lines 3 and 4).

Finally, we note that the Decision Adapter model can learn to implement the same function as cGate [13], but is theoretically more powerful. While cGate uses a linear elementwise product between the context and state features, our model uses a hypernetwork to generate the weights of a nonlinear adapter module. Our hypernetwork is general enough to be able to recover this elementwise product, but is not constrained to do so. See Appendix C for a more formal treatment of this point.

# 6 Experimental Setup

## 6.1 Metrics

Our evaluation strategy is as follows: We take the trained model, and compute the average of the total episode reward over $n = 5$ episodes on each evaluation context $c$ to obtain a collection of contexts $C$ and corresponding rewards $R$. We then calculate the **Average Evaluation Reward (AER)** : $AER = \frac{1}{c_{max} - c_{min}} \int_{c_{min}}^{c_{max}} R(c)dc$, where $c_{min}$ and $c_{max}$ are the minimum and maximum context values in $C$ respectively, and $R(c)$ is the reward obtained on context $c$. This metric is high when the agent performs well across a wide range of contexts, and thus corresponds to generalisation performance. However, since we generally have $\mathcal{C}_{train} \subset \mathcal{C}_{evaluation}$, this metric also incorporates how well the agent performs on the training contexts.[4] Despite this, it is still a useful metric for generalisation, as our training context set typically contains only a handful of contexts, with most of the evaluation contexts being unseen.

## 6.2 Baselines

We use Soft-Actor-Critic [39] for all methods, to isolate and fairly compare the network architectures. We aim to have an equal number of learnable parameters for each method and adjust the number of hidden nodes to achieve this. We use the following baselines, with more details in Appendix D:

**Unaware** This model simply ignores the contextual information and just uses the state. This approach allows us to evaluate how well a method that does not incorporate context performs.

**Concat** This method forms a new, augmented state space $\mathcal{S}' = \mathcal{S} \times \mathcal{C}$, and then gives the model the concatenation $[s; c]$ of the state and context [16–18, 37]. This baseline allows us to compare our method to the current standard approach of using contextual information.

**cGate** cGate [13] learns separate state ($\phi(s)$) and context ($g(c)$) encoders, and predicts the action $a = f(\phi(s) \odot g(c))$, where $f$ is the learned policy and $\odot$ is the elementwise product.

**FLAP** FLAP [42] learns a shared state representation across tasks, which is then processed by a task-specific linear layer that is generated by conditioning on the context.

Our Adapter configuration is more fully described in Appendix D.3, and Appendix E contains ablation experiments comparing the performance of different hyperparameter settings.

## 6.3 Environments

### 6.3.1 ODE

This environment is described by an ordinary differential equation (ODE), parametrised by $n$ variables making up the context. The dynamics equation is $x_{t+1} = x_t + \dot{x}_t dt$, with $\dot{x} = c_0 a + c_1 a^2 + c_2 a^3 + \ldots$, truncated at some $c_{n-1}$. The episode terminates after 200 timesteps, with the reward function:

---

[4]The AER can be computed over only the unseen evaluation contexts, but we include the training contexts to obtain a more general measure of performance. However, the difference between the two versions is minor, see Appendix F.4.

$$R_t = \begin{cases} 1 & \text{if } |x| < 0.05 \\ \frac{1}{4} & \text{if } |x| < 0.5 \end{cases} \qquad \begin{cases} \frac{1}{2} & \text{if } |x| < 0.1 \\ \frac{1}{20} & \text{if } |x| < 2 \end{cases} \qquad \begin{cases} \frac{1}{3} & \text{if } |x| < 0.2 \\ 0 & \text{otherwise} \end{cases}$$

This incentivises the agent to give control $a$ to keep the state close to $x = 0$. The context in this environment is $c = [c_0, \ldots, c_{n-1}] \in \mathbb{R}^n$. The action space is two-dimensional, continuous and bounded, i.e., $a_0, a_1 \in [-1, 1]$. The action is interpreted as a complex number, $a = a_0 + a_1 i$, to allow the dynamics equation to be solvable, even in normally unsolvable cases such as $\dot{x} = a^2$. Here, restricting $a$ to $\mathbb{R}$ would result in $\dot{x} \geq 0$ and an unsolvable system for initial conditions $x_0 > 0$. We keep only the real part of the updated state and clip it to always fall between $-20$ and $20$.

The ODE acts as a conceptually simple environment where we can arbitrarily scale both the number (i.e., $n$) and magnitude (i.e., $|c_i|$) of the contexts and precisely measure the effects. Furthermore, since many dynamical systems can be modelled using differential equations [83–85], the ODE can be considered a distilled version of these. Finally, context is necessary to perform well in this environment, making it a good benchmark. See Appendix D for more details.

### 6.3.2 CartPole

CartPole [86] is a task where an agent must control the movement of a cart to balance a pole vertically placed upon it. The observation space is a 4-dimensional real vector containing the cart's position $x$ and velocity $\dot{x}$, as well as the pole's angle $\theta$ and angular velocity $\dot{\theta}$. We use a continuous action space where $a \in [-1, 1]$ corresponds to the force applied to the cart (where negative values push the cart to the left and positive values to the right). The reward function is $+1$ for each step that the pole is upright. An episode terminates when the pole is tilted too far off-center, the cart's position is outside the allowable bounds or the number of timesteps is greater than $500$. We follow prior work and consider the variables `Gravity`, `Cart Mass`, `Pole Length`, `Pole Mass` and `Force Magnitude` collectively as the context [13], even if only a subset of variables change. Here, we change only the pole length and evaluate how susceptible each model is to distractor variables. In particular, during training, we also add $k$ additional dimensions to the context, each with a constant value of $1$. Then, during evaluation, we set these values to $0$. This may occur during a real-world example, where accurately identifying or inferring the pertinent context variables may be challenging [18, 87], and lead to several irrelevant variables. Overall, this means that the context is represented as a $5 + k$-dimensional vector. This environment is useful as it (1) is a simple setting with models that can be trained quickly, and (2) still has desirable aspects, namely a way to perturb some underlying variables to change the dynamics of the environment. Furthermore, prior work has shown that even standard, unaware RL algorithms can generalise well in this environment [88] (in the absence of distractor context variables), which allows us to investigate the benefits of context and detriment of distractors.

### 6.3.3 Mujoco Ant

As a more complex and high-dimensional problem, we consider *Ant* from the Mujoco suite of environments [89, 90]. Here, the task is to control a 4-legged robot such that it walks. The observation space consists of the robot's joint angles and velocities, as well as contact forces and torques applied to each link, totalling $111$ dimensions. The action space $\mathcal{A} = [-1, 1]^8$ represents the torque applied to each of the 8 joints. Furthermore, each episode is terminated after $1000$ timesteps, with a positive reward for not having fallen over at each step and moving forward. The reward function also penalises large control forces. Here, the mass is the context variable, which is $5$ by default. We train on the set $\{5, 35, 75\}$ and evaluate on 200 evenly spaced points between $0.5$ and $100$.

## 7 Results

### 7.1 Generalisation Performance

In this section, we consider the ODE environment, with a training context set of $\{-5, -1, 1, 5\}$ and 300k training timesteps. To measure generalisation, we use an evaluation context range consisting of 201 equally spaced points between $-10$ and $10$ (inclusive). In the left pane of Fig. 3, we plot the average performance across the entire evaluation range (as discussed in Section 6.1) at different points during training.

Firstly, we can see that the Unaware model fails to generalise well, since the optimal actions in the same state for two different contexts may be completely different. This result substantiates our theoretical analysis in Section 4, highlighting the importance of using context in this domain. Concat, `cGate` and FLAP perform reasonably well, but our Decision Adapter outperforms all methods and converges rapidly. In Appendix F, we plot separate subsets of the evaluation range: **Train**, **Interpolation** and **Extrapolation**, corresponding to the contexts in the training set, the contexts within the region $[-5, 5]$ and the contexts outside the convex hull of the training contexts, containing $c \in [-10, -5) \cup (5, 10]$, respectively. There we find that most methods, with the exception of the Unaware model, perform well on the training contexts, and the Adapter outperforms all methods when extrapolating. These results also extend to the multidimensional context case, shown in Fig. 3 (right). We train on the context set given by $\{1, 0, -1\}^2 \cup \{5, 0, -5\}^2$, with the context $(0, 0)$ being omitted due to it being unsolvable.[5] The evaluation range consists of the cartesian product $A^2$, where $A$ contains 21 equally-spaced points between $-10$ and $10$. The Adapter outperforms all methods, with the Concat model coming second. FLAP and the Unaware models struggle in this case.

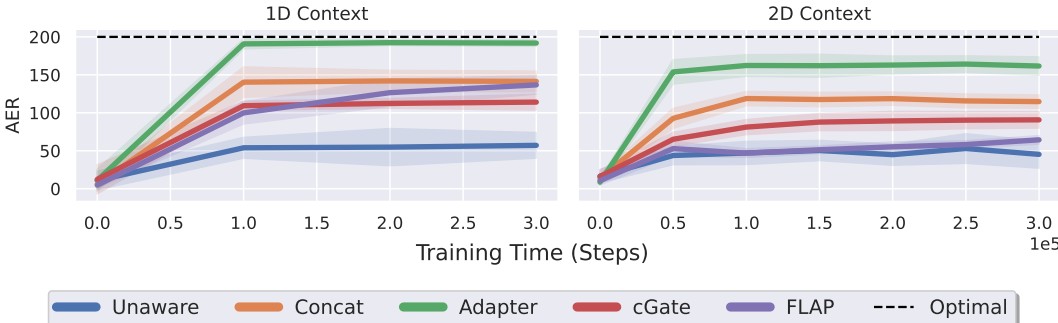

Figure 3: The average reward over the evaluation range at various points during training for the ODE domain. Mean performance is shown, with standard deviation (across 16 seeds) shaded. The left pane shows the one-dimensional context results, while the right pane shows the two-dimensional results.

## 7.2 Robustness to Distractors

For this and the next section, we consider the Concat and `cGate` models as baselines. The reasons for this are that (1) the Concat model performed comparably to or better than the other baselines in the previous sections, making it a representative example of the context-aware approaches; and (2) the Concat model is currently a very prevalent way of incorporating context [16–18, 35–38]; and (3) `cGate` is the closest approach to our work in the current literature.

The results for CartPole are shown in Fig. 4. When there are no distractor variables, the Adapter, Concat and `cGate` models perform comparably. In this case, each architecture achieves near the maximum possible reward for the domain. Further, we see little effect after adding just one confounding context variable. However, as we add additional distractor variables, the Concat and `cGate` models' performances drop significantly, whereas the Adapter's performance remains relatively stable. Strikingly, the Adapter architecture trained with 100 distractor context variables is still able to perform well on this domain and significantly outperforms the Concat model with significantly fewer (just 20) distractor variables. This demonstrates that, given only useful context, concatenating state and context is a reasonable approach in some environments. However, this is a strong assumption in many practical cases where we may be uncertain about which context variables are necessary. In such cases, the consequences of ignoring the conceptual differences between state and context are catastrophic and it is necessary to use the Adapter architecture. We find a similar result in the ODE, which we show in Appendix F.5.

As an extension to the previous experiment, here we consider using non-fixed distractor variables, i.e., sampling them from Gaussian distribution with different means. The intuition behind this experiment is similar to before, but allows for small variations within training and testing. In particular, we use $\sigma = 0.2$ and a mean of either 0 or 1. In this experiment, a new distractor context is sampled at the start of every episode. During training this is sampled from $\mathcal{N}(1, 0.2)$ and during evaluation we

---

[5]Here, for a set $S$, $S^2 = S \times S = \{(a, b) | a, b \in S\}$ denotes the cartesian product of $S$ with itself.

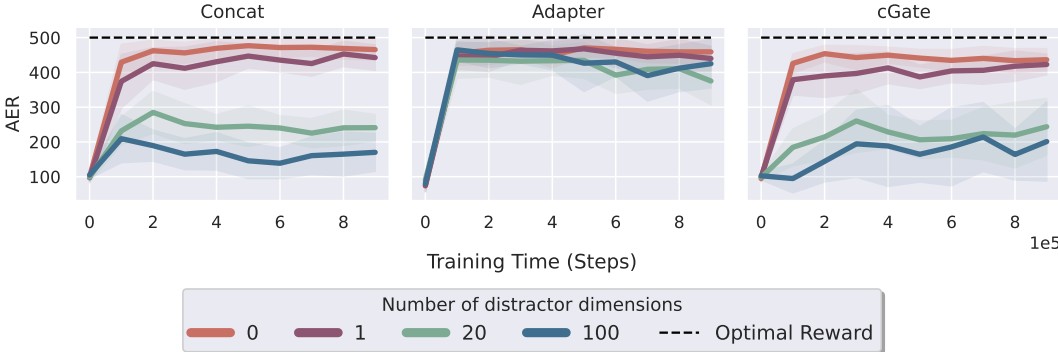

Figure 4: Showing the sensitivity of the (left) Concat, (middle) Adapter and (right) `cGate` models to distractor context variables in CartPole. The mean and standard deviation over 16 seeds are shown.

sample from $\mathcal{N}(0, 0.2)$. In CartPole, the results in Fig. 5 show that the Adapter is still more robust to changing distractor variables compared to either Concat or `cGate`. See Appendix F.6 for more details, results on the ODE domain, and additional experiments when the mean of the Gaussian does not change between training and testing.

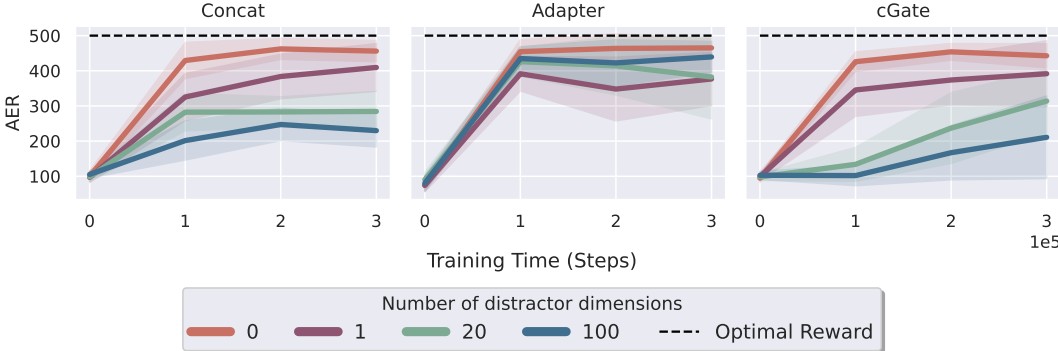

Figure 5: The sensitivity of Concat, Adapter and `cGate` to distractor context variables—sampled from a Gaussian distribution—in CartPole. We show the mean and standard deviation over 16 seeds.

### 7.3 High-Dimensional Control

We next consider a more complex and challenging continuous-control environment, that of the Mujoco Ant. We run the same experiment as in CartPole, where we add distractor variables, with no effect on the dynamics, and different values between training and testing. These results are shown in Fig. 6. Overall, we observe a similar result to that of CartPole—the Concat and `cGate` models are significantly less robust to having irrelevant distractor variables. By contrast, the Adapter consistently performs well regardless of how many distractor dimensions are added.

## 8 Limitations

While our Adapter model performs well empirically, it does have some limitations, which we summarise here and expand on in Appendix G. First, we find that using an incorrect and noisy context (Appendix G.1) generally resulted in the Adapter model performing worse, particularly as we increased the level of noise. However, the Adapter model still performs comparably to the Concat model in this case. Second, we generally normalise the contexts before we pass them to the models, relative to the maximum context encountered during training. While our Adapter performs well when the context normalisation is incorrect by a factor of 2 or 3, its performance does suffer when the normalisation value is orders of magnitudes too small, leading to very large contexts being input into the model (see Appendix G.2 for further discussion). Third, when training on a context range that is too narrow (see Appendix G.3) or does not exhibit sufficient variation, our model is susceptible to overfitting, which

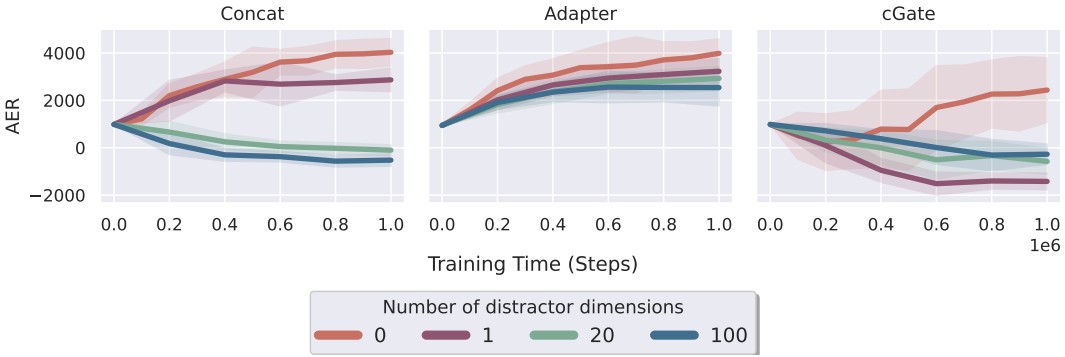

Figure 6: Showing the sensitivity of the (left) Concat, (middle) Adapter and (right) `cGate` models to distractor context variables in Ant. The mean and standard deviation over 16 seeds are shown.

leads to worse generalisation. Even if our model does not overfit, an overly narrow context range would also lead to poor generalisation. Thus, ensuring that agents are trained in sufficiently diverse environments is still important. Fourth, while we assumed access to the ground-truth context, it may not always be available in practice [31]. Therefore, integrating the Decision Adapter into existing context inference methods [18, 33] is a promising avenue for future work. A final limitation of our Adapter is its longer wall-clock training time compared to other methods (given the same training timesteps and a similar number of parameters). This is likely caused by the hypernetwork that we use. However, we did not utilise strategies such as caching the generated weights for a particular context (which would be possible since context remains constant for an episode), or more effectively batching the forward passes through the Adapter which could ameliorate this issue in practice.

Beyond addressing these limitations, there are several logical directions for future work. First, we only consider state and context features to be numerical vectors; therefore, extending our approach to different modalities—for instance, image observations and natural language context—would be a natural extension. In principle, our Adapter model could be directly applied in this setting without any fundamental modifications. One option would be to explore adding the hypernetwork directly into the modality-specific model (e.g., a convolutional neural network in the case of images), and the alternative would be to integrate the adapter module in the final (fully-connected) layers of the RL policy. Second, while we focused on the problem of zero-shot generalisation, our approach may also be suitable for efficient few-shot learning. In particular, adapters in NLP are often added to a large pre-trained model and then fine-tuned while freezing the primary model [76]. Using the Decision Adapter in this paradigm is a promising direction for future work, especially in cases where the difference between training and testing is so large that we cannot expect effective zero-shot generalisation. Combining our work with fine-tuning-based [91–93] or meta-learning [36, 60, 68, 69, 94–96] approaches, which generally focus more on this case, could likewise be promising.

## 9 Conclusion

In this work, we consider the problem of generalising to new transition dynamics. We first illustrate that for some classes of problems, a context-unaware policy cannot perform well over the entire problem space—necessitating the use of context. We next turned our attention to how context should best be incorporated, introducing the Decision Adapter—a method that generates the weights of an adapter module based on the context. When context is necessary, our Decision Adapter outperforms all of our baselines, including the prevalent Concat model. Next, when there are some irrelevant context variables that change between training and testing, the Concat and `cGate` models fail to generalise, whereas our Decision Adapter performs well. This result holds in several domains, including the more complex Ant. Furthermore, our Adapter is theoretically more powerful and empirically more effective than `cGate`, showing that our hypernetwork-based perspective is useful when dealing with context. Ultimately, we find that the Decision Adapter is a natural, and **robust**, approach to reliably incorporate context into RL, in a factored way. This appears to be of particular benefit when there are several irrelevant context variables, for instance, in real-world settings [97].

## Acknowledgements

Computations were performed using High Performance Computing infrastructure provided by the Mathematical Sciences Support unit at the University of the Witwatersrand. M.B. is supported by the Rhodes Trust. D.J. is a Google PhD Fellow and Commonwealth Scholar. B.R. is a CIFAR Azrieli Global Scholar in the Learning in Machines & Brains program.

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

# Appendix

## A    Formalising the Theoretical Foundations

In this section, we more formally consider our theoretical results from Section 4. We first define our problem formulation in Assumption A.1, then state the theorem in Theorem A.5. Finally, we prove the theorem at the end of this section.

**Assumption A.1. (Problem Formulation):** Suppose we have a CMDP where $\mathcal{S}, \mathcal{A} \subseteq \mathbb{R}^d$ and $\gamma \in (0, 1)$. The context $c \in \mathbb{R}^d$ defines a goal location, with a reward function $R_c(s, a, s') = 1$ if $||s' - c|| < \bar{\tau}$ and 0 otherwise. The episode terminates when the reward obtained is 1. Suppose further that $\bar{\tau} = \tau D$, with $D \in \mathbb{R}^+$ being the distance the agent travels in a single step and $\tau \in \mathbb{Z}^+$ being the number of steps it takes the agent to traverse a distance of $\bar{\tau}$. Here $\tau$ and $D$ are kept fixed across contexts.

**Definition A.2. (Context Averaged Value Function):** Denote $\bar{V}_{\mathcal{C}}^{\pi}(s) = \mathbb{E}_{c \sim \mathcal{C}}[V_c^{\pi}(s)]$, i.e., the expected value of the value function of policy $\pi$ in state $s$, over all contexts in the set $\mathcal{C}$. Similarly $\bar{V}_{\mathcal{C}}^{*}(s) = \mathbb{E}_{c \sim \mathcal{C}}[V_c^{*}(s)]$ is the average of the value functions of the optimal policies on each context.

**Definition A.3. (Value Function Optimality Ratio):** Let $0 \leq \alpha_{\mathcal{C}}^{\pi}(s) = \frac{\bar{V}_{\mathcal{C}}^{\pi}(s)}{\bar{V}_{\mathcal{C}}^{*}(s)} \leq 1$ denote how close the average value function of policy $\pi$ is to the optimal context-specific value function. We use the optimality ratio instead of the optimality gap [47] because the optimality gap depends heavily on the reward scale of the environment, whereas the optimality ratio depends only on the ratios.

**Definition A.4. (Minimum Inter-context Distance):** Let $d_{min} = \min_{c_i \neq c_j \in \mathcal{C}} \lceil \frac{||c_i - c_j||}{D} \rceil$ be the minimum number of steps required to travel between any two context centers.

**Theorem A.5.** *Then, in this problem space, we have the following:*

    (i) *For some set of contexts $\mathcal{C}_{far}$ with cardinality $N$, and some state $s$, all deterministic unaware policies $\pi$ will have:*

$$\alpha_{\mathcal{C}_{far}}^{\pi}(s) = \frac{\bar{V}_{\mathcal{C}_{far}}^{\pi}(s)}{\bar{V}_{\mathcal{C}_{far}}^{*}(s)} \leq \frac{1}{N} \frac{1 - \gamma^{N d_{min}}}{1 - \gamma^{d_{min}}} \tag{1}$$

    *Additionally:*

$$\lim_{d_{min} \to \infty} \alpha_{\mathcal{C}_{far}}^{\pi}(s) \to \frac{1}{N} \tag{2}$$

    *i.e., as the contexts within $\mathcal{C}_{far}$ move further away from each other the agent's performance scales inversely with the number of contexts.*

    (ii) *For some set, $\mathcal{C}_{close}$ and all states $s$, at least one deterministic unaware policy $\pi_g$ will have:*

$$\frac{\bar{V}_{\mathcal{C}_{close}}^{\pi_g}(s)}{\bar{V}_{\mathcal{C}_{close}}^{*}(s)} > \gamma^{\tau} \tag{3}$$

    *regardless of the number of contexts in $\mathcal{C}_{close}$.*

*Proof.* (i) Consider a set of contexts $\mathcal{C}_{far}$ such that for all $c_i, c_j \in \mathcal{C}_{far}, i \neq j, ||c_i - c_j|| > 4\bar{\tau}$. Consider the state $s = \frac{1}{N} \sum_{i=1}^{N} c_i$, i.e., in the middle of all contexts. Now, suppose the optimal policy per context would require only $\beta$ steps to reach the appropriate goal. An unaware policy that reaches every goal, by contrast, must travel to each context in sequence. If this policy visits $c_1, c_2, \dots, c_N$, in order, then the value functions for each context would be:

- $V_{c_1}^{\pi}(s) = \gamma^{\beta} \cdot 1 = V_{c_1}^{*}(s)$

- $V_{c_2}^{\pi}(s) = \gamma^{\beta + \lceil \frac{||c_1 - c_2||}{D} \rceil}$

- $V_{c_3}^{\pi}(s) = \gamma^{\beta + \lceil \frac{||c_1 - c_2||}{D} \rceil + \lceil \frac{||c_2 - c_3||}{D} \rceil}$,

and so on. Let $d_{min}$ be the number of steps required to move between the two closest distinct contexts in $\mathcal{C}_{far}$. Thus, $d_{min} \geq \frac{4\bar{\tau}-\bar{\tau}-\bar{\tau}}{D} = 2\tau.$[6] Then, $V_{c_i}^\pi(s) \leq \gamma^{\beta+(i-1)d_{min}}$. Now,

$$\bar{V}_{\mathcal{C}_{far}}^\pi(s) = \mathbb{E}_{c\sim\mathcal{C}_{far}}[V_c^\pi(s)]$$

$$= \frac{1}{N}\sum_{i=1}^N V_{c_i}^\pi(s)$$

$$\leq \frac{1}{N}\sum_{i=1}^N \gamma^{\beta+(i-1)d_{min}}$$

$$= \gamma^\beta \frac{1}{N}\sum_{i=1}^N \gamma^{(i-1)d_{min}}$$

$$= \gamma^\beta \frac{1}{N}\frac{1-\gamma^{Nd_{min}}}{1-\gamma^{d_{min}}}$$

$$= \bar{V}_{\mathcal{C}_{far}}^*(s)\frac{1}{N}\frac{1-\gamma^{Nd_{min}}}{1-\gamma^{d_{min}}}$$

Hence, $\alpha = \frac{\bar{V}_{\mathcal{C}_{far}}^\pi(s)}{\bar{V}_{\mathcal{C}_{far}}^*(s)} \leq \frac{1}{N}\frac{1-\gamma^{Nd_{min}}}{1-\gamma^{d_{min}}}$. Finally, as $d_{min} \to \infty$, $\gamma^{d_{min}} \to 0$ and $\gamma^{Nd_{min}} \to 0$. Hence, $\lim_{N\to\infty} \alpha = \frac{1}{N}$.

(ii) Consider a set of contexts $\mathcal{C}_{close}$ such that for all $c_i, c_j \in \mathcal{C}_{close}$, $||c_i - c_j|| < \bar{\tau}$. Then, consider an unaware policy that always travels to the joint intersection point of these contexts. This intersection point is at most $\bar{\tau}$ away from the closest context to the agent, therefore the agent will perform at most $\tau$ unnecessary steps compared to the optimal policy. Hence, its value function has a lower bound of $V_{c_i}^*(s)\gamma^\tau$, and we have

$$V_{c_i}^*(s) \geq V_{c_i}^\pi(s) \geq V_{c_i}^*(s)\gamma^\tau$$

Therefore,

$$\bar{V}_{\mathcal{C}_{close}}^\pi(s) = \mathbb{E}_{c\sim\mathcal{C}_{close}}[V_c^\pi(s)]$$

$$\geq \mathbb{E}_{c\sim\mathcal{C}_{close}}[V_c^*(s)\gamma^\tau]$$

$$= \gamma^\tau \mathbb{E}_{c\sim\mathcal{C}_{close}}[V_c^*(s)]$$

$$= \gamma^\tau \bar{V}_{\mathcal{C}_{close}}^*(s)$$

And we have: $\frac{\bar{V}_{\mathcal{C}_{close}}^\pi(s)}{\bar{V}_{\mathcal{C}_{close}}^*(s)} \geq \gamma^\tau$ $\qquad\square$

# B  Linking Theory and Practice

The formulation in Assumption A.1 has consistent dynamics and context-dependent rewards (as the context defines the goal location, and the transition dynamics do not depend on the context). This is in contrast to the problem we aim to study, which has consistent rewards, but context-dependent dynamics; however, we can show that for the environment considered in Appendix A, these two are equivalent.

We can obtain the context-dependent dynamics formulation under the assumptions in Section 2, i.e., that only the transition dynamics change, as follows. Suppose the context defines a rotation matrix $C_i \in \mathbb{R}^{d\times d}$, with dynamics equation $T(s, a) = s' = s + C_i a$ and initial state $s_0 = 0$. Here the goal is a fixed location $g \in \mathbb{R}^d$. The episode would terminate with a reward of 1 if $||s' - g|| < \bar{\tau}$. We can then perform a coordinate transform to obtain $C_i^{-1}s' = C_i^{-1}s + a$. The goal would then depend on the context $g_{c_i} = C_i^{-1}g$ and the effect of an action in this new space would be consistent across contexts.

---

[6]This is because, while the centers of the circles $c_i$ and $c_j$ are further than $4\bar{\tau}$ apart, their closest edges are only at least $2\bar{\tau}$ apart.

In essence, we change the reference frame to be that of the agent, leading to separate goal locations for each context, but consistent dynamics—corresponding to the problem in Assumption A.1.

In the remainder of this section, we link our theoretical and empirical results, showing that the ODE (for some context distributions) is an example of a non-overlapping environment, while in CartPole, an unaware policy can perform well.

## B.1 ODE

Here we show that, in the ODE domain, the Unaware model can either perform well or poorly, depending on the context set it is evaluated on. In particular, we train models on context sets $\mathcal{C}_{train}^{A} = \{1, 5\}$ and $\mathcal{C}_{train}^{B} = \{-5, -1, 1, 5\}$ and evaluate on $\mathcal{C}_{test}^{A} = [0, 10]$ and $\mathcal{C}_{test}^{B} = [-10, 10]$. These results are shown in Fig. 7. When only considering the positive contexts (left), the Unaware model performs only slightly worse than the Concat model. This corresponds to case (ii) in Theorem A.5, where the contexts are similar enough for a single policy to perform well. However, when considering both positive and negative contexts (right), the Unaware model fails completely, and performs much worse than the context-aware ones. This is because, in the ODE domain, positive and negative contexts cannot be solved using the same action. This corresponds to case (i) in Theorem A.5, where making progress on one context results in negative progress on another.

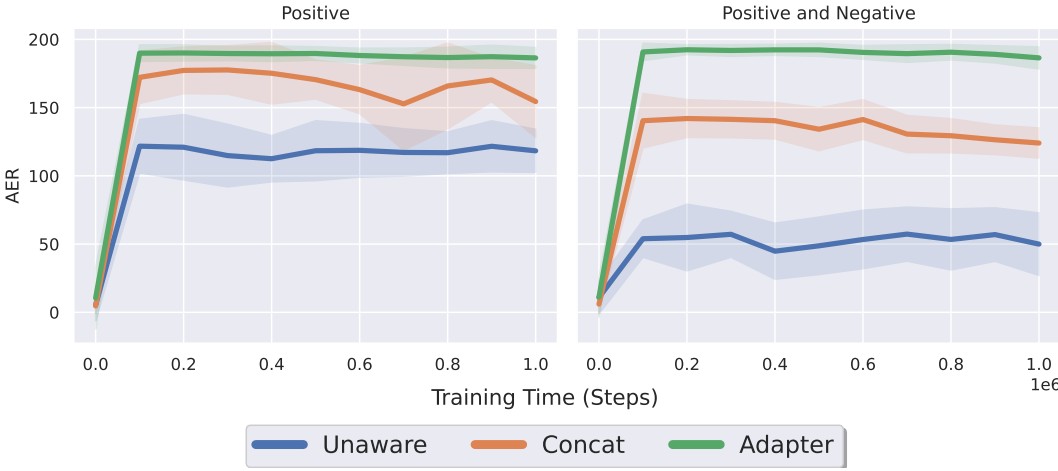

Figure 7: Here we show results in the ODE domain. On the left, we train on $\{1, 5\}$ and evaluate on $[0, 10]$. On the right, we consider both negative and positive contexts, training on $\{-5, -1, 1, 5\}$ and evaluating on $[-10, 10]$. The Unaware model performs much better when only considering positive contexts, even though the context-aware models still outperform it.

## B.2 CartPole

We now investigate CartPole. In particular, we vary only the pole length, and consider the training contexts as $\{1, 4, 6\}$, with the evaluation contexts being 301 equally-spaced points between 0.1 and 10.

These results are shown in Fig. 8. While the Unaware model can achieve comparable generalisation performance to the context-informed models, it takes significantly longer to do so. In particular, to obtain an average reward above 400, the Unaware model must train for more than 600k steps, whereas `cGate`, Concat and our Adapter reach this threshold before 100k steps. This shows that, while context is not necessary in this domain, incorporating it may be beneficial and can lead to much more sample-efficient generalisation compared to the Unaware model. This is particularly relevant when training on multiple training contexts, which seems to cause interference for the Unaware model. The context-aware approaches, however, are not as susceptible to this problem, as they are able to distinguish experience from different contexts. Furthermore, most of the context-aware models perform similarly in this case. In particular, our Adapter, Concat and `cGate` converge rapidly, while FLAP converges slightly slower, but achieves comparable generalisation performance at the end of training.

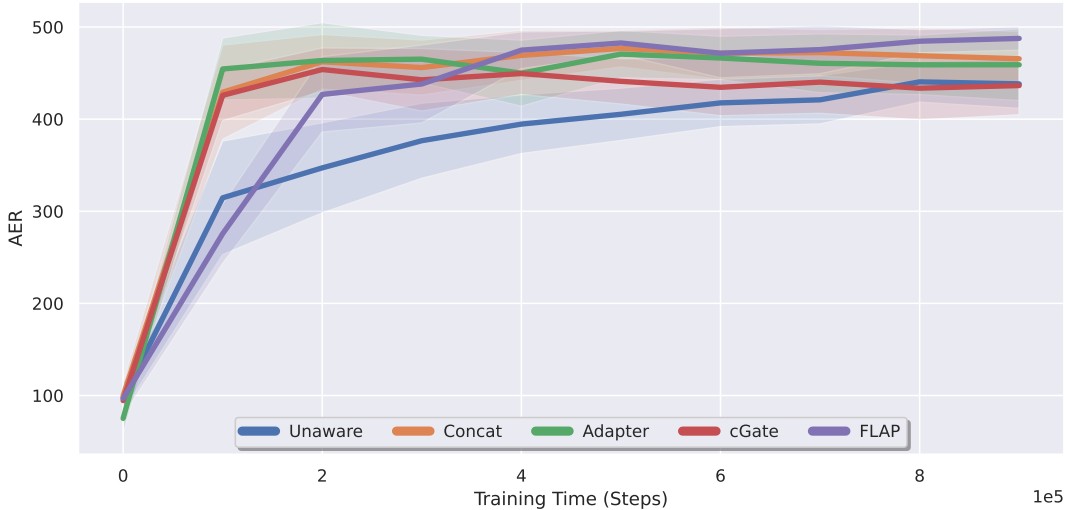

Figure 8: Showing the average evaluation reward as we increase training time for CartPole. Mean performance over seeds is shown and standard deviation is shaded.

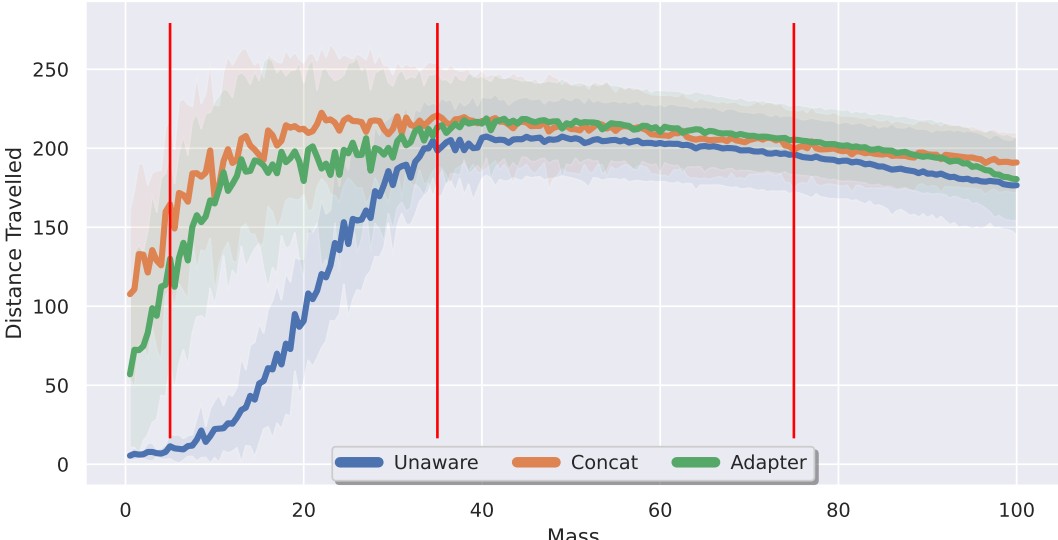

Figure 9: Mass vs. Distance Travelled (proportional to reward), at the end of training for the Ant domain. Red lines are the training contexts. Mean performance is shown, with standard deviation over 16 seeds shaded.

### B.3 Ant

Here we consider the Ant domain, and vary the mass, without any distractor variables. As seen in Fig. 9, the Unaware model performs worse than the context-aware ones. This performance gap is due to the Unaware model being unable to simultaneously perform well on light and heavy masses, as seen in Fig. 9. In contrast, by utilising context, the agent can perform well in both settings despite their significant differences.

## C cGate as a Special Case of the Decision Adapter

As discussed in Section 5, here we show that our architecture is, in fact, a generalisation of another network architecture introduced by prior work. Benjamins et al. [13] introduce `cGate`, which obtains an action as follows. It first calculates $\phi(s) \in \mathbb{R}^d$ using the state encoder $\phi$ and $g(c) \in \mathbb{R}^d$ using the

context encoder $g$. Then, it calculates features $h = \texttt{cGate}(s, c) = \phi(s) \odot g(c)$, with $\odot$ being the elementwise product. The action is obtained as $a = f(h)$, where $f : \mathbb{R}^d \to \mathcal{A}$ is the learned policy. Here, both the state encoder $\phi$ and the context encoder $g$ correspond to neural networks that each output a $d$-dimensional vector.

This approach is a specific instantiation of our general adapter architecture, where we have one adapter module $A = A_i$. In particular, the state encoder $\phi$ corresponds to the partial neural network before our adapter module, i.e., layers $L_1, L_2, \ldots, L_{i-1}$. The policy $f$ corresponds to the layers after our adapter module, $L_i, L_{i+1}, \ldots, L_n$. Thus, we have that $\phi(s) = L_{i-1}(x_{i-1})$ and $f(h) = L_n(x_n)$. If we define our adapter network $A(\phi(s)|c) = \phi(s) \odot g(c)$, we can recover the same result as $\texttt{cGate}$. This would be possible if we let our adapter consist of one layer without a nonlinearity or a bias vector, with its weight matrix being set to $W = H(c) = diag(g(c))$. This would result in

$$A(\phi(s)|c) = W\phi(s) = diag(g(c))\phi(s) = g(c) \odot \phi(s) = \texttt{cGate}(s, c),$$

where we have used the fact that, for two vectors $a, b \in \mathbb{R}^d$, the matrix multiplication $diag(a)b$ is the same as the elementwise product $a \odot b$. This output vector is then passed to the policy network $f$. Our approach, however, is strictly more powerful than a single elementwise product, as our adapter can be an entire nonlinear neural network.

# D  Experimental Details

## D.1  Environments

This section contains more details about the experimental setup of our environments.

### D.1.1  Context Normalisation

Before we pass the ground truth context to the models, we first normalise them to a range between $0$ and $1$. In particular, for each context dimension $i$, we calculate the context to be given to the agents $c_{norm,i}$ as

$$c_{norm,i} = \frac{c_i}{max_i},$$

where $max_i$ is the largest value for dimension $i$ in the training context set. This results in the largest training contexts having values of $1.0$. During evaluation, we perform the same normalisation, so contexts outside the convex hull of the training set will potentially result in values larger than $1.0$.

### D.1.2  ODE

During training, for each context, we cycle between starting states, with each episode taking a new starting state from the set $\{1.0, 0.5, -0.5, -1.0\}$. Thus, the agent trains on the first context for 4 episodes, each with a different initial state. Then it goes on to the second context, and so on. When we reach the end of the context list, we simply start at the first context again. We do this to ensure training is comparable for each method, and that there is sufficient diversity in the initial state distribution. During evaluation, we fix the starting state at $x_0 = 1.0$ to isolate the effect of context.

### D.1.3  CartPole

Table 1 illustrates the context variables we use in CartPole. While we consider all five variables as the context, we varied only the pole length for our experiments. The value for each state variable at the start of the episode is randomly sampled from the interval $[-0.05, 0.05]$.

Table 1: The five context variables and their default values.

| Name | Symbol | Default Value |
|---|---|---|
| Gravity | $g$ | 9.80 |
| Cart Mass | $m_c$ | 1.00 |
| Pole Mass | $m_p$ | 0.10 |
| Pole Length | $l$ | 0.50 |
| Force Magnitude | $\tilde{F}$ | 10.00 |

## D.2 Hyperparameters

We use `ReLU` for all activation functions. We also aim to have an equal number of learnable parameters for each method and adjust the number of hidden nodes to achieve this.

We use Soft-Actor-Critic [39, SAC] for all methods, to isolate and fairly compare the network architectures. We use the high-performing and standard implementation of SAC from the *CleanRL* library [98] with neural networks being written in PyTorch [99]. We use the default hyperparameters, which are listed in Table 2. SAC learns both an actor and a critic. The actor takes as input the state and outputs $\mu$ and $\log(\sigma)$, which are used to construct the probability distribution over actions $\mathcal{N}(\mu, \sigma)$ which is sampled from to obtain an action. The critic receives the performed action $a$ as well as the state $s$, and outputs a single number, representing the approximate action-value $\hat{q}(s, a)$. Fig. 10 illustrates each baseline.

Table 2: The default hyperparameters we use in the CleanRL implementation.

| Name | Value |
|---|---|
| Buffer Size | 1 000 000 |
| $\gamma$ | 0.99 |
| $\tau$ | 0.005 |
| Batch Size | 256 |
| Exploration Noise | 0.1 |
| First Learning Timestep | 5000 |
| Policy Learning Rate | 0.0003 |
| Critic Learning Rate | 0.001 |
| Policy Update Frequency | 2 |
| Target Network Update Frequency | 1 |
| Noise Clip | 0.5 |
| Automatically Tune Entropy | Yes |

## D.3 Adapter Configuration and Design Decisions

Now, our method provides a great degree of flexibility in terms of configuration, and we must make multiple design decisions to obtain a usable instantiation of the Decision Adapter. In this section, we briefly detail the default settings we use and justify why these are reasonable. We empirically investigate the merits of these choices and compare them against alternative options in Appendix E.

### D.3.1 Bottleneck Architecture

We use a bottleneck architecture in our generated adapter modules, as described by Houlsby et al. [76]. In essence, we have a down-projection layer that transforms features from $d_i$-dimensional to $p$-dimensional, with $p < d_i$. Then, the second layer up-projects this back to a $d_i$-dimensional vector.

The bottleneck architecture is beneficial for two reasons. Firstly, it reduces the number of required parameters in our adapter model. For instance, if we have two layers in this bottleneck architecture, the number of parameters would differ substantially compared to one layer that transforms the $d_i$-dimensional features to $d_i$ dimensions. In the first case, the number of parameters would be on the order of $O(d_i \times p + p \times d_i)$, corresponding to a $d_i \times p$ matrix for the down-projection and a $p \times d_i$ matrix for the up-projection.[7] If we only have one layer, then we have a single $d_i \times d_i$ matrix. Thus, the bottleneck requires $O(pd_i)$ parameters, whereas the single layer requires $O(d_i^2)$. If we, for example, have $d_i = 256$ and $p = 32$, the bottleneck architecture has four times fewer parameters than the single layer. This, in turn, makes it easier for the adapter hypernetwork to generate useful weights.

Secondly, the bottleneck architecture imparts a low-rank inductive bias on the adapter, which may prevent it from overfitting [100]. This idea is related to prior work that has shown that, counterintuitively, reducing the capacity of the agent's neural network can improve generalisation and reduce

---

[7]The bias vectors correspond to an additional $p + d_i$ parameters, but we omit this term as it is dominated by the number of parameters in the weight matrices.

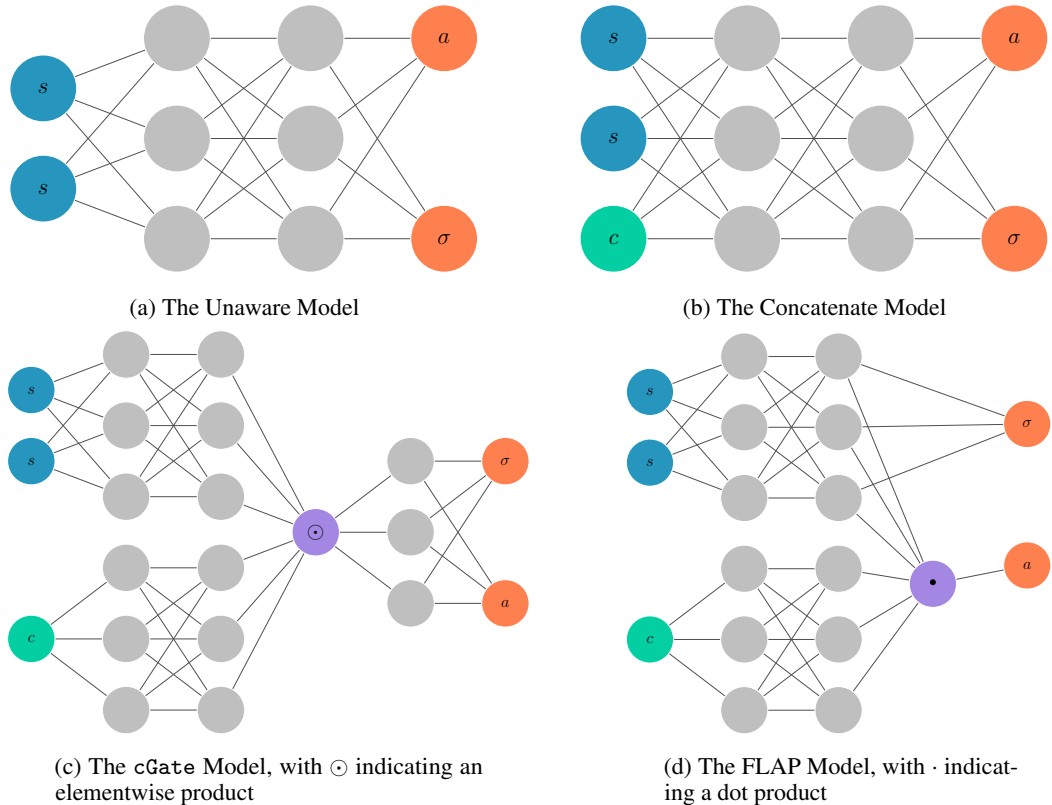

(a) The Unaware Model

(b) The Concatenate Model

(c) The `cGate` Model, with ⊙ indicating an elementwise product

(d) The FLAP Model, with · indicating a dot product

Figure 10: Illustrating the architectures of each of our baselines. Here we consider the actor's architecture, with two separate output heads, one predicting the mean action $a$ and the other predicting the log of the standard deviation (denoted as $\sigma$ in this figure). In each plot, the blue nodes represent the state inputs whereas the green one represents the context. In general, for the bottom row, the network architectures for the context and state encoders are not necessarily the same.

overfitting [101–103]. Although our approach does not incorporate information-bottleneck losses like these methods, the observed benefits of reduced capacity provide additional intuition for utilising the bottleneck architecture. Finally, despite any possible information loss due to the bottleneck layer, the skip connection allows the network to retain any important information from the input.

### D.4 Compute

For compute, we used an internal cluster consisting of nodes with NVIDIA RTX 3090 GPUs. For a single experiment and a single method, the runs took between 1 and 3 days to complete all seeds.

## E Adapter Ablations

Here we experimentally justify some of the adapter-specific design decisions we made for our empirical results. The experiments in this section also give us insight into how robust the Decision Adapter is to changing its configuration options. We investigate the effects of the following factors, with our conclusions listed in **bold**.

- The network architecture of the Adapter module. We find that **Most adapter architectures perform well.** This result is expanded upon in Appendix E.1.
- Whether the hypernetwork uses chunking. **Hypernetwork chunking outperforms non-chunking methods while using significantly fewer parameters** (Appendix E.2).
- Using a skip connection. **The Decision Adapter can perform well with or without a skip connection** (Appendix E.3).

- The location of the adapter module in the main network. **Most locations lead to high performance, except if the adapter is at the very start or very end of the network** (Appendix E.4).

- Having an activation function before the adapter. **Not having an activation function before the adapter in the actor model outperforms the alternatives** (Appendix E.5).

For this entire section, we consider the ODE environment and follow the same procedure as in Section 7.1. We use the ODE due to its fast training, and the fact that context is necessary, which enables us to determine if a particular design choice of the Adapter results in poor use of the context.

## E.1 Adapter Architecture

Here we examine the effects of changing the Adapter module's network architecture. We are interested in how the architecture impacts performance for two reasons. First, we wish to empirically justify our bottleneck architecture. Second, we wish to show that the Adapter performs well with a wide variety of network architectures, and is not particularly sensitive to this choice.

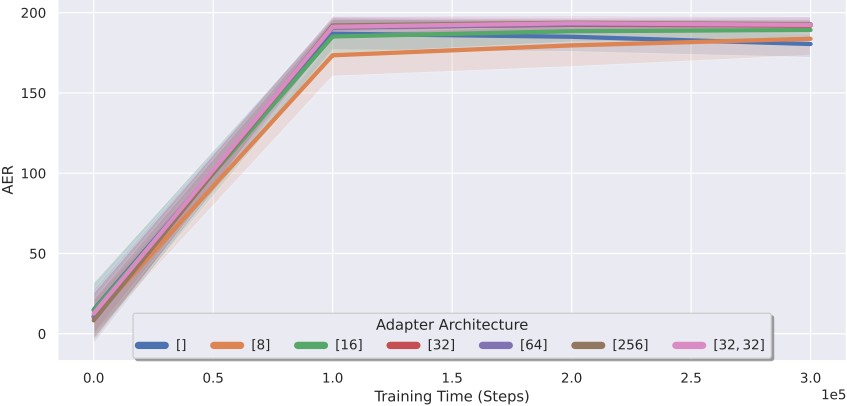

Figure 11: Evaluating the effect of different adapter architectures. Mean performance is shown, with standard deviation shaded.

As discussed in Appendix D.3, we use a bottleneck architecture of size 32. Thus, the 256-dimensional features are transformed into 32-dimensional features and then projected back to 256 dimensions. We denote this architecture as [32], since the adapter has a single hidden layer of 32 neurons. Fig. 11 illustrates the performance for various other adapter architectures. Most methods perform comparably, but using 8 hidden nodes or not having any hidden layers performs slightly worse than the base architecture. A large hidden layer of 256 nodes also does not outperform our bottleneck architecture. Finally, there is no particular benefit to having multiple layers in the adapter module.

## E.2 Hypernetwork Chunking

Next, we compare our default architecture choice, which uses hypernetwork chunking, against non-chunking settings. We do this because hypernetwork chunking allows us to have significantly fewer learnable parameters (resulting in faster training) compared to the alternative of predicting the entire weight vector in one forward pass. Thus, if our approach performs comparably or better than non-chunking settings, then the chunking architecture is justified.

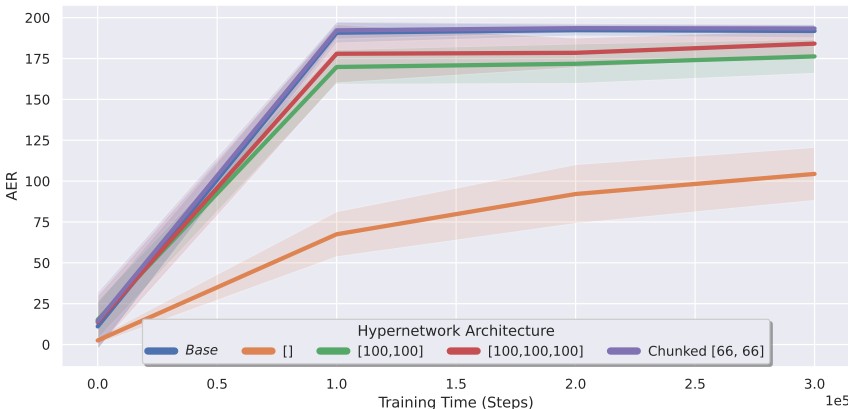

Figure 12: Comparing hypernetwork chunking ($Base$) compared to various non-chunking configurations as well as using larger chunks (Chunked[66, 66]). Mean performance is shown, with standard deviation shaded. Here, the values in the brackets represent the number of hidden nodes per layer, with [ ] corresponding to a hypernetwork that has no hidden layers.

We consider a non-chunking hypernetwork architecture of no hidden layers, as well as 2 and 3 hidden layers of size 100, respectively. Our final option is a chunked hypernetwork, with chunks of size 660, and two hidden layers of size 66 each (this is roughly twice as large as our default configuration option discussed in Appendix D.3). These results are shown in Fig. 12. We find that both chunking configurations perform the best. The non-chunking approaches perform slightly worse, particularly if the hypernetwork has no hidden layers.

These non-chunking configurations also have significantly more learnable parameters than the default. For instance, the base setting results in the actor's hypernetwork having around 16k parameters, whereas the [100, 100], non-chunking setting corresponds to roughly 1.7M parameters. In summary, by using hypernetwork chunking, we can obtain high performance while using a very small number of learnable parameters.

### E.3 Skip Connection

Here we investigate whether a skip connection is beneficial or not. As mentioned in Section 5, the skip connection sets the updated $x$ as $x = A(x) + x$, whereas if we remove it, the function becomes just $x = A(x)$. In Fig. 13, we can see that using a skip connection performs similarly to the alternative. This shows that the Decision Adapter is robust to this parameter, and can perform well with or without a skip connection.

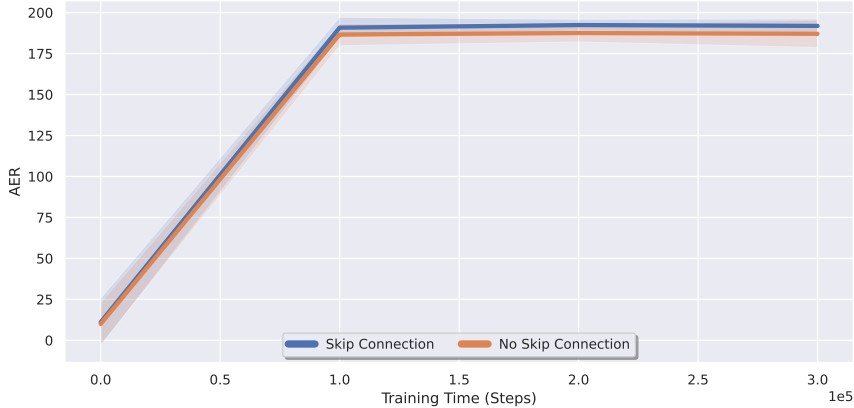

Figure 13: Evaluating the effect of using a skip connection. Mean performance is shown, with standard deviation shaded.

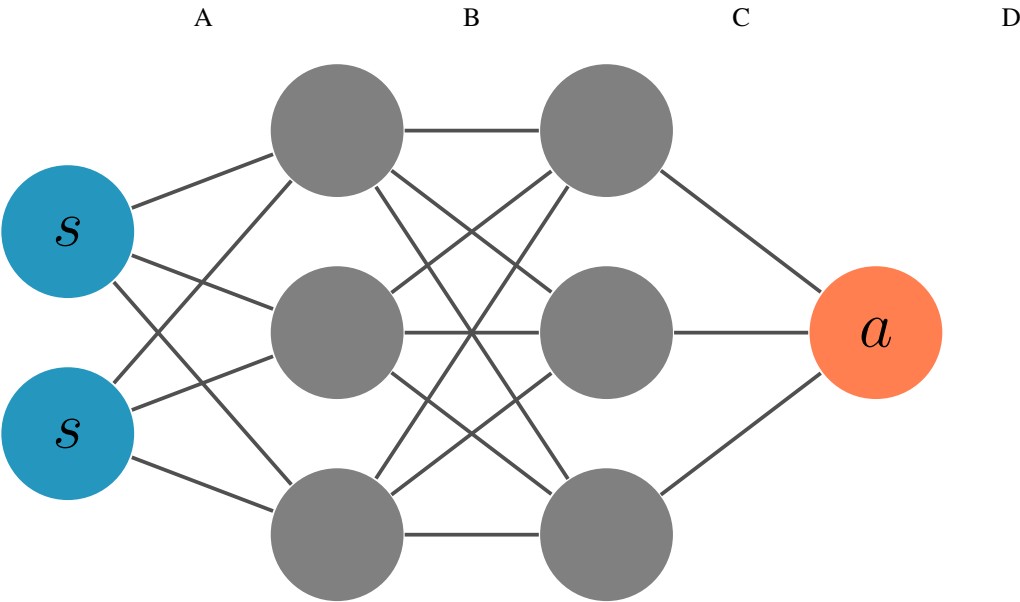

Figure 14: Showing the base network architecture, with letters indicating adapter positions. For instance, an adapter placed at location (D) receives the action as input and returns a modified one.

## E.4 Adapter Location

We next examine the effects of changing the location of the adapter module in our network, and what happens if we have multiple modules. This information would be useful when using our method in practice, and deciding where and how many modules to place. There are generally many options regarding the location, as an adapter module can be placed between any two layers of the primary network.

As described in Section 5, our actor network consists of a "trunk", containing one hidden layer, which maps the state to a 256-dimensional feature vector, and a one-layer action head, directly mapping the trunk features to an action. Here we consider the following locations for the adapter in both the actor and critic networks:

**Base** The standard adapter used in the rest of this chapter, which is placed before the action head's layer. In the critic, the adapter is placed before the last layer. This corresponds to location (C) in Fig. 14.

**Base$_F$** Placing the adapter in the actor network before the final layer of the trunk, leaving the critic unchanged (location B).

**Start** The adapter here is at the start of both networks, i.e., the adapter module receives the state as input (location A).

**End** Here, we place the adapter at the very end. Thus, the adapter takes in the predicted action and outputs a modified one. The critic's adapter is also placed at the end. This corresponds to location (D) in Fig. 14.

**All$_F$** Three adapter modules, one at the start of the trunk, one before and one after its final layer. The critic also has three adapters, one at the start, one before the last layer and one after the last layer. The actor has adapters in locations (A), (B) and (C).

In all cases, to ensure comparability against the base model, there is no activation function after the trunk in the actor model. These results are illustrated in Fig. 15. Overall, the base model performs the best. The adapter in the trunk performs slightly worse and the performance is quite poor if we put the adapter modules at the start of the networks. If we have three adapter modules, the performance is similar to only having one before the final layer. Finally, when adapting only the final action, the performance is also poor. Overall, the performance is low if we have the adapter only at the very

start or end of the network. Within a range of locations in the "middle" of the network, however, the performance is high and only slightly worse than the default setting.

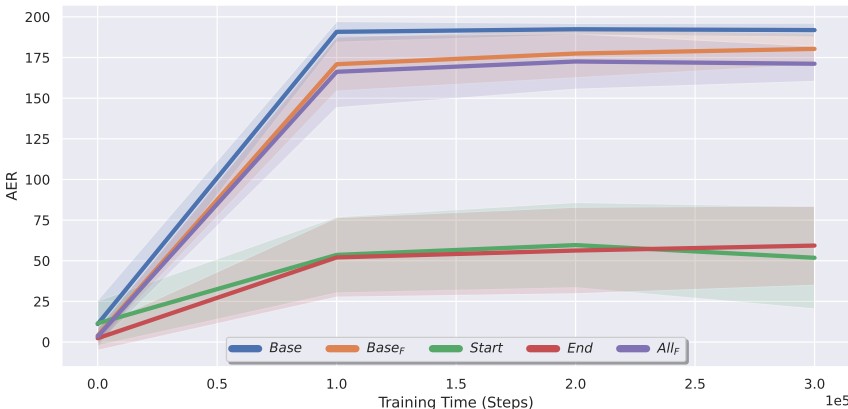

Figure 15: Evaluating the effect of the adapter module's location. Mean performance is shown, with standard deviation shaded.

### E.5 Activation Function Before Adapter

Our final ablation considers the benefits of not having an activation function before the adapter module inside the actor network. We perform this experiment to determine if our choice of not having an activation function before the adapter in the actor is justified, and if it is necessary to do the same for the critic.

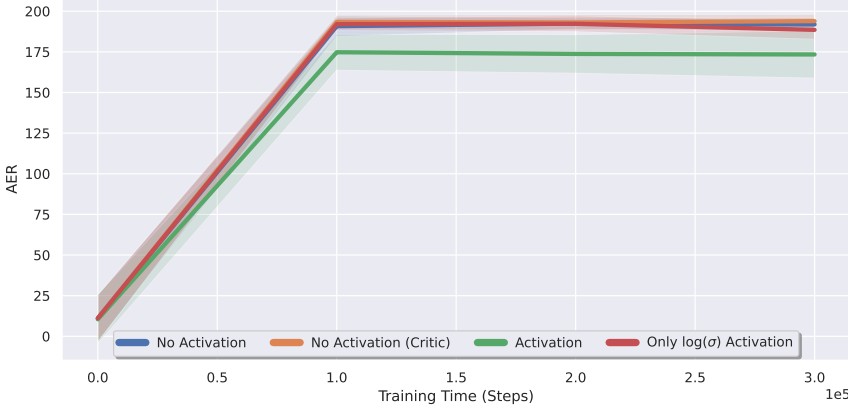

Figure 16: Comparing the effect of having an activation function before the adapter vs. not. The blue, green and red lines alter the actor's architecture, whereas orange changes the critic. Blue corresponds to the base choice outlined in Appendix D.3. Mean performance is shown, with standard deviation shaded.

Here we consider four options: (1) not having an activation function after the trunk; (2) having one; (3) having an activation function after the trunk, but only for the standard deviation head; and (4) not having an activation before the adapter in the critic. Option (1) corresponds to the default choice. These results are shown in Fig. 16. Not having an activation function outperforms having one, and having an activation for only the standard deviation head is similar to not having one. Thus, not having an activation function before the adapter inside the *critic* network performs as well as the base option, which has this activation. Overall, this result justifies our design choice by showing that it performs comparably to or better than the alternatives.

# F   Additional Results

This section contains some additional results that we refer to in the main text.

## F.1   Single Dimension Interpolation, Train & Extrapolation

See Fig. 17 for the performance of each model on the ODE on the particular context sets, Interpolation, Training and Extrapolation. These plots therefore correspond to subsets of the evaluation range in the left pane of Fig. 3.

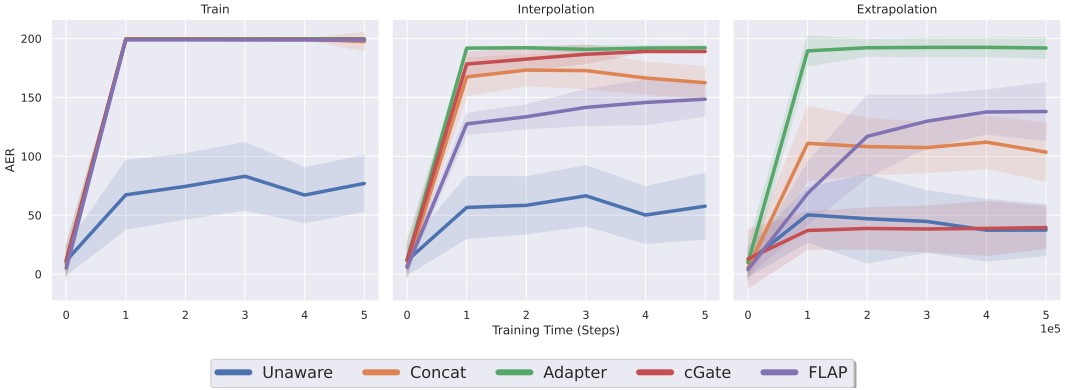

Figure 17: Showing the performance of each method in three different evaluation regimes, (left) the training contexts, (center) interpolation and (right) extrapolation. Mean performance is shown, with standard deviation shaded.

## F.2   Multiple Dimension Heatmaps

Fig. 18 illustrates the granular performance of each method in the ODE domain, corresponding to the right pane of Fig. 3. Our Decision Adapter outperforms all baselines and, as can be seen in Figs. 18a and 18b, generalises further away from the training contexts than the Concat model. In particular, these heatmaps show that both models perform well on the training contexts (the blue squares). However, the Adapter generalises much better, especially to contexts far outside the training range. For example, when $|c_1| \in [8, 10]$ (corresponding to the far left and far right of the heatmaps), the Adapter's rewards are higher than the Concat model's. The Adapter obtains between 50 and 150 more reward than the Concat model – which is a substantial difference, as the maximum reward in this environment is 200. This region of context-space is far outside of the convex hull of the training range, which contains values of $|c_1|$ only up to 5.0. Closer to the training range; for instance, around the boundary of the green square, the Adapter still outperforms the Concat model, but the difference is less pronounced.

FLAP and `cGate` perform poorly, even though they obtain near-perfect rewards on the training contexts. There may be several reasons for this. First, here we train on 16 relatively sparsely distributed contexts, with 300k steps in total, resulting in fewer episodes in each context compared to Section 7.1. Second, with two dimensions, the interpolation range makes up a smaller proportion of the evaluation range compared to the one-dimensional case. This is because, in one dimension, the range $[-5, 5]$ makes up half of the $[-10, 10]$ evaluation range. In two dimensions, however, the interpolation range makes up only a quarter of the evaluation set. Thus, the overall average performance in Fig. 3 is more skewed towards extrapolation. This, coupled with the fact that both `cGate` and FLAP perform poorly on extrapolation (Fig. 17, right), may explain why they underperform in this case. Lastly, it is generally easier to generalise in the one-dimensional ODE, as the optimal action depends mainly on the signs of the context and state variables. By contrast, the multidimensional case is more difficult, as the dynamics equation includes a nonlinear action term. This is also why, for instance, our Decision Adapter performs worse in this case (around 160 overall evaluation reward) compared to the one-dimensional ODE (around 190).

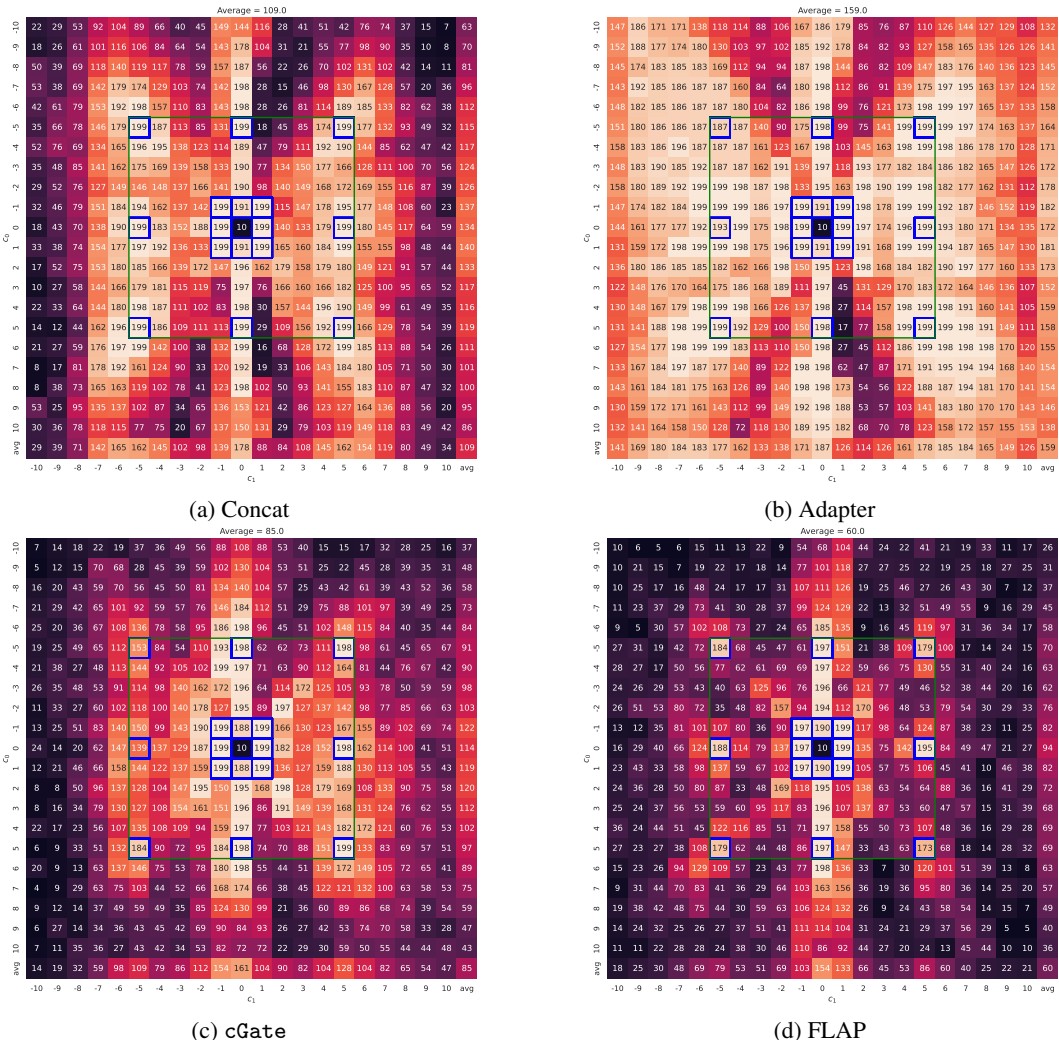

Figure 18: Multidimensional contexts in the ODE domain for (a) Concat, (b) Adapter, (c) `cGate` and (d) FLAP. Heatmap Performance for (a) `cGate` and (b) FLAP. Each cell represents the reward obtained when evaluated in the corresponding context, averaged over 16 seeds. The blue squares indicate the specific training contexts, whereas the green square represents the convex hull of training contexts. For the heatmaps, the last rows and columns, labelled *avg*, represent the average reward over the particular column or row, respectively.

## F.3 Additional Baselines

We consider two ablations as additional baselines. The first, called `AdapterNoHnet`, is based on our adapter module, but there is no hypernetwork involved. The adapter's architecture and location are the same, but it is now a single MLP that takes in the concatenation of the state-based features and the raw context (as the model has to be context-conditioned, and must also process the state-based features). The second baseline, termed `cGateEveryLayer`, uses the same elementwise operation as `cGate`, but this happens at every hidden layer except at only one. Fig. 19 illustrate these results on the 1D and 2D ODE domains respectively, with the Adapter, Concat and `cGate` models there for reference. Overall, `cGateEveryLayer` does not outperform `cGate`, and `AdapterNoHnet` performs much worse than our hypernetwork-based adapter.

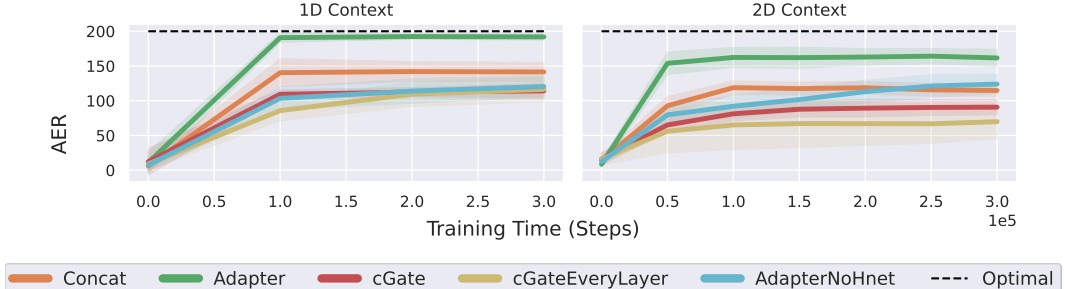

Figure 19: The performance on the 1D and 2D ODE domains (i.e., similar to Fig. 3) of the additional baselines. The mean and standard deviation over 16 seeds are shown.

## F.4 AER Metric

Fig. 20 illustrates the difference between using the AER metric as defined in Section 6.1 (corresponding to Fig. 20a) and *only* considering the testing contexts (corresponding to Fig. 20b). Overall, the performance of each model is very similar; this is because the training contexts make up a small proportion of the entire evaluation range.

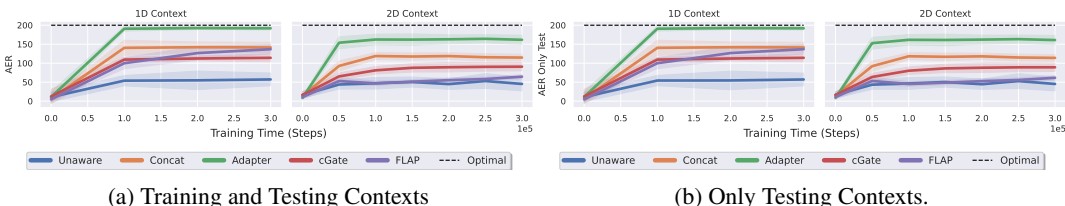

(a) Training and Testing Contexts  (b) Only Testing Contexts.

Figure 20: Comparing (a) averaging performance over all contexts and (b) restricting the AER metric to only unseen testing contexts.

## F.5 Distractors

The ODE domain displays similar distractor results to our other environments (see Section 7.2). In particular, Fig. 21 shows that the Adapter is robust to adding more irrelevant distractor variables, whereas Concat and `cGate` are not.

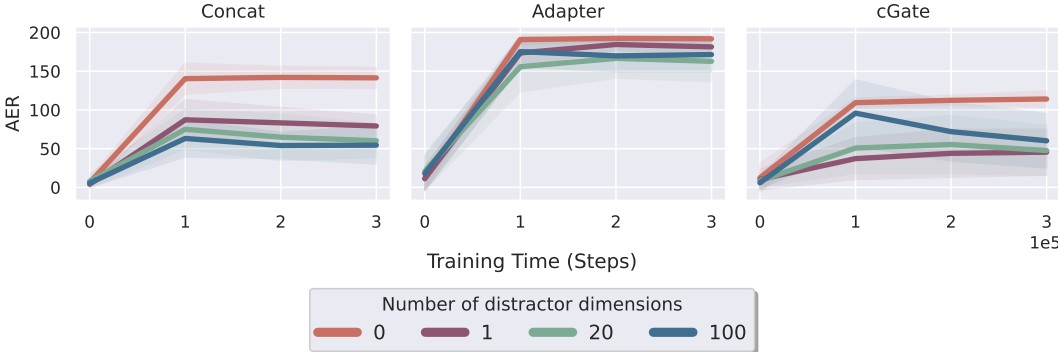

Figure 21: Showing the sensitivity of the (left) Concat, (middle) Adapter and (right) `cGate` models to fixed distractor context variables on the ODE. The mean and standard deviation over 16 seeds are shown.

## F.6 Gaussian Distractor Experiments

This section expands upon the results in Section 7.2 and contains additional results where the distractor variables are not fixed.

### F.6.1 Difference between training and testing

Fig. 5 contains the results on CartPole, and the ODE results are shown in Fig. 22. For both domains, the conclusion is similar in that the Adapter is still more robust to changing distractor variables compared to either Concat or `cGate`.

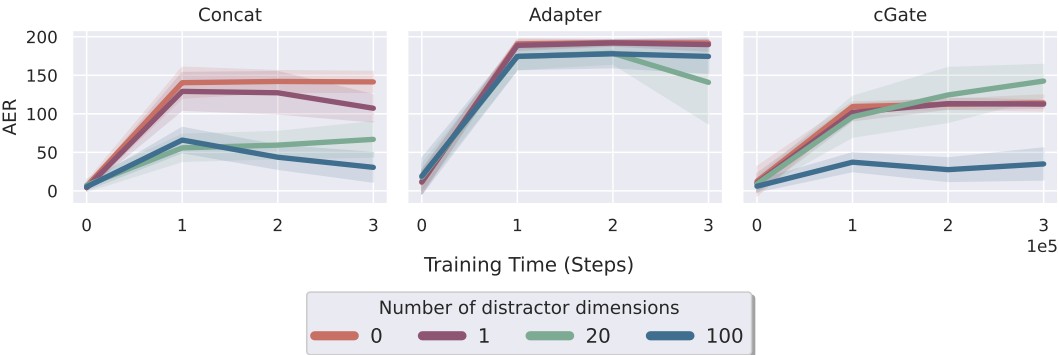

Figure 22: The Gaussian distractor variable results. This corresponds to using a mean of 1 during training and a mean of 0 during testing. We use a consistent $\sigma = 0.2$. These results are for the ODE domain, showing the mean and standard over 16 seeds.

### F.6.2 Keeping the training and testing distractor distributions the same

Next, we consider a similar case, but the means of the Gaussians are the same across training and testing. Overall, the results for the ODE and CartPole (with a mean of 0 during both training and testing) are shown in Fig. 23. For the ODE, while the Adapter is insensitive to adding more distractor dimensions, both `cGate` and the Concat perform worse as we add more irrelevant distractor dimensions to the context. However, the difference is much less pronounced than in the case where the means are different between training and testing. Surprisingly, in the ODE, `cGate` performs slightly better with 20 distractor variables than with 0. These variables may effectively act as an additional, noisy, bias term that the model can utilise. For CartPole, the Concat and `cGate` models perform much worse with 100 distractor dimensions compared to none. The Adapter displays some noise, indicating that some seeds performed poorly while others performed well. This is similar to the results we observed in Appendix G.1—too much noise leads to unstable training.

Finally, we consider using a mean of 1 during both training and testing in Fig. 24. For the ODE, each model performs similarly regardless of the number of distractor variables. In CartPole there is not a

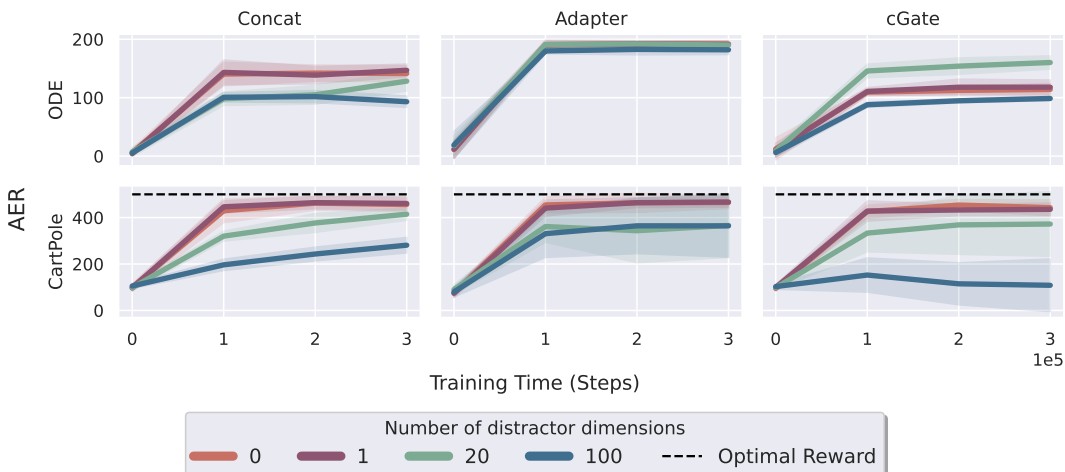

Figure 23: The Gaussian distractor variable results, using a consistent mean of 0 during training and testing, with $\sigma = 0.2$. These results are for the (top) ODE and (bottom) CartPole domains, showing the mean and standard over 16 seeds.

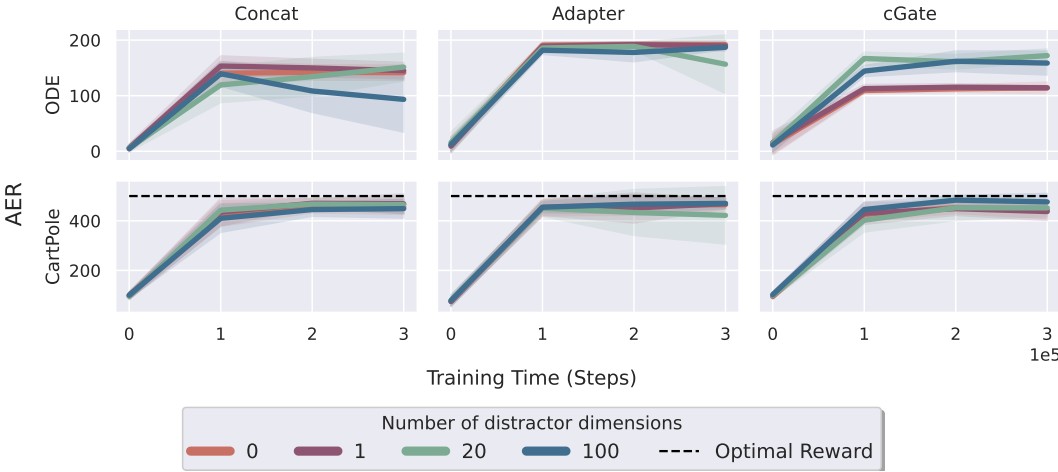

Figure 24: The Gaussian distractor variable results, using a consistent mean of 1 during training and testing, with $\sigma = 0.2$. These results are for the (top) ODE and (bottom) CartPole domains, showing the mean and standard over 16 seeds.

large difference between the various models when the mean of the distractor variables is 1 during both training and testing.

### F.6.3  Summary

Overall, The Adapter is significantly less susceptible to overfitting to irrelevant distractors compared to the Concat model and `cGate`. However, if these distractors remain the same during training and testing (which may not be particularly likely), it matters less whether the model overfits or not, and all models are less affected by adding more distractors.

## G  Limitations

This section expands upon the limitations of our approach, outlined in Section 8.

### G.1  Noisy Contexts

Now, while we use the ground truth context to isolate the effect of the network architecture on generalisation performance, this is not always available [12, 18]. A common approach is to learn a context encoder that uses experience in the environment to infer an approximation to the context, which is then used to act [18, 87]. This encoded context, however, may not always be entirely accurate. Thus, in this experiment, we examine how sensitive each context-aware model is to receiving noisy contexts during evaluation. We consider two cases; the first is where the models received the uncorrupted context during training, and the second is where the models also encountered noise during training. The first case could reasonably occur in reality, since we may have access to the ground truth context during training, but not evaluation [14]. The second case, where we train with noise, could correspond to training the RL policy alongside the context encoder, leading to some noise during training. This case additionally allows us to examine whether training with noise increases the models' robustness to context noise during evaluation.

We take the models trained during the primary CartPole experiments, in Appendix B.2, where we varied the pole length. We use the model checkpoints that were trained for 600k timesteps, as both the Concat and Adapter models did not improve much by training for longer than this (see Fig. 8). The noise we add here, corresponding to normally distributed noise with varying standard deviations, was added to each context dimension instead of just the one representing pole length. We experiment with a standard deviation $\sigma \in \{0, 0.05, 0.1, 0.2, 0.5, 1\}$. The noise was added after normalising the contexts such that the largest training context had a value of $1.0$. Thus, a standard deviation of $1.0$ represents a substantial amount of noise. In addition to the models trained without noise, we

additionally train models that encountered noise *during* training. In particular, we consider standard deviations $\sigma \in \{0.1, 0.5\}$, and change no other aspect of training.

The results are illustrated in Fig. 25. The left pane shows the performance of the models trained without noise. Overall, these results show that the Concat model is more susceptible to noisy contexts than our Adapter, bolstering the argument that treating context as a state dimension is not ideal [13]. Both models, however, generalise worse as the context becomes more inaccurate. This is in contrast to the Unaware model, which performs equally well regardless of the level of context noise. The Adapter still outperforms the Unaware model until the noise level reaches around $\sigma = 0.5$. This noise level, however, is very large, and $\sigma = 0.5$ corresponds to a pole length of 3 meters. The default pole length is $0.5$ meters, and the largest training length is 6 meters.

When training with a small amount of noise (Fig. 25, middle) both models perform similarly to training without noise. However, when training with a large amount of noise, corresponding to the right pane of Fig. 25, the Concat model becomes more robust to noisy contexts, but the Adapter performs worse. In particular, the Adapter model exhibits a large amount of variation across seeds, indicating that some seeds performed badly. On average, the Adapter performs similarly to the Unaware model until the level of context noise increases beyond the training noise of $\sigma = 0.5$.

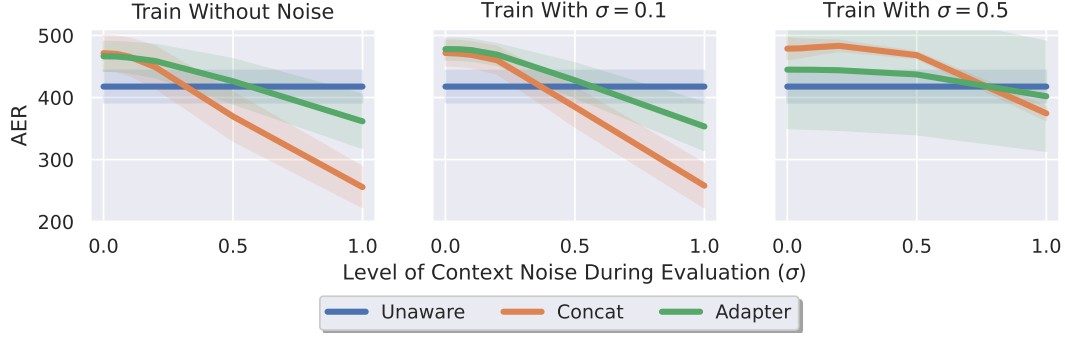

Figure 25: Comparing the average evaluation reward (after 600k steps) as a function of the standard deviation $\sigma$ of the Gaussian noise added to the context during evaluation. In the left pane, the models trained without noise are shown. The middle pane corresponds to training with $\sigma = 0.1$ and the right pane shows the results when training with $\sigma = 0.5$. Mean performance is shown, with standard deviation over 16 seeds shaded.

## G.2 Suboptimal Context Normalisations

### G.2.1 ODE

As discussed in the main text, we first normalise the context before passing it to the agents. This is generally done by using the largest absolute value encountered during training. For instance, for the one-dimensional ODE experiment, we used $c_{norm} = \frac{c-0}{5}$ as the normalisation function. Here we determine how sensitive the Decision Adapter is to changes in this procedure. This is useful as we may often need to choose the normalisation factor without knowing exactly which variations we will encounter during evaluation. To do this, we run four additional experiments, with normalisation values of $0.1$, $1.0$, $2.0$ and $15.0$, respectively. We train the agents for only 300k steps, as after training for this number of steps, most models performed similarly to their final performance. The results are shown in Fig. 26, and we can see that the Concat and Adapter models are both robust to the normalisation value. Our Decision Adapter loses less performance than the Concat model as we change the normalisation value. Despite losing some performance as the context normalisation value increases, both context-aware models still outperform the Unaware model.

### G.2.2 CartPole

We now consider a similar setup in CartPole. In particular, we train on a Pole Mass of $0.1$. We choose this variable as some of our experiments showed that the Unaware model generalises equally well regardless of the training contexts. We consider two scenarios: In the first, we normalise with respect to the `Small` setting (i.e. $c_{norm} = \frac{c}{0.01}$). Thus, during training, the normalised context had a value

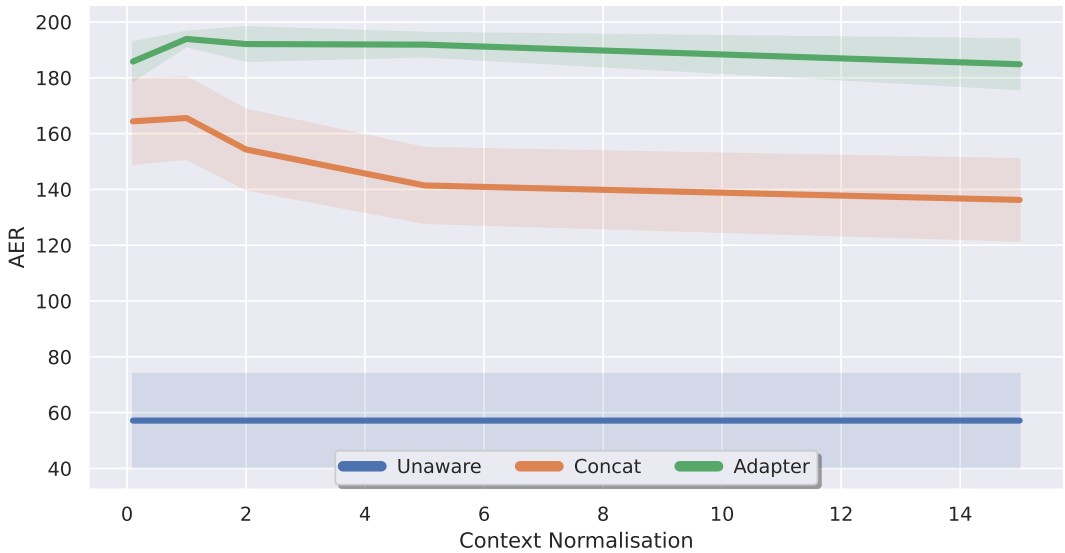

Figure 26: Showing the overall performance (y-axis) of each context-aware model after 300k steps as we change the context normalisation value (x-axis). Mean performance is shown, with standard deviation shaded.

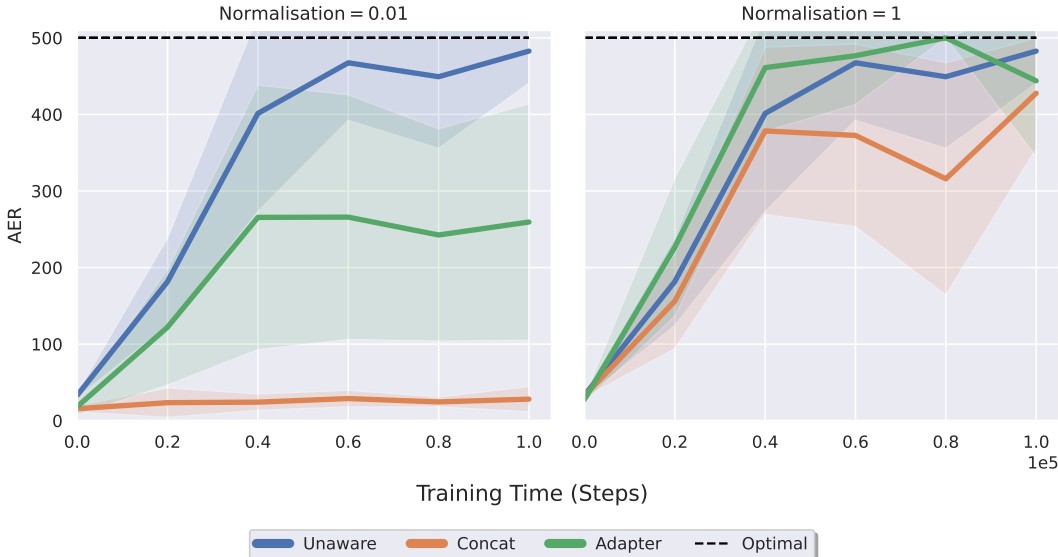

Figure 27: Performance when evaluating on changing Pole Mass when normalising with respect to (left) Mass = 0.01 and (right) Mass = 1.0. In both plots, the models trained on Mass = 0.01. Mean performance is shown, with standard deviation shaded.

of $1.0$; during evaluation, it was nearly always larger than $1.0$. In the second case, we normalise with respect to the X-Large setting (corresponding to $c_{norm} = \frac{c}{1}$). We trained on the Small pole length for both settings. Fig. 27 shows the results for this experiment. When normalising with respect to the Small setting, the Concat model fails to generalise at all, and the Adapter performs worse than Unaware. When changing the normalisation, however, the context-aware models perform significantly better.

### G.2.3   Summary

Thus, the Decision Adapter is relatively robust to small changes in context normalisations. However, when encountering evaluation contexts that are 100 times larger than during training, appropriate

normalisation is crucial and can make a drastic difference – even when training on the exact same contexts. As seen in the ODE, having a normalisation value that is slightly too large does not cause significant performance penalties. Thus, normalising the contexts with a larger value than expected during training is a good heuristic.

## G.3 Narrow Context Range

We next consider the effect of changing the set of training contexts. Here we aim to briefly illustrate that the training context range can have a large effect on the performance of the agents. In particular, we consider the ODE domain and 8 separate sets of training contexts. In all of these cases, we keep the normalisation consistent at $c_{norm} = \frac{c}{5}$ to ensure comparability. The results when training on each context set are shown in Fig. 28. Overall, when we have a single positive and negative training context, performance is poor when this context is very small (a), and increases as it becomes larger (d and f). When we have multiple contexts, but insufficient variation (b, c, e), then performance is also suboptimal. For instance, in (b), the training contexts cover only the small region $[-1, 1]$. In (c), the training set of contexts does not contain any negative contexts, leading to poor generalisation. Finally, if contexts are varied, but spread out too far (g), the performance also suffers.

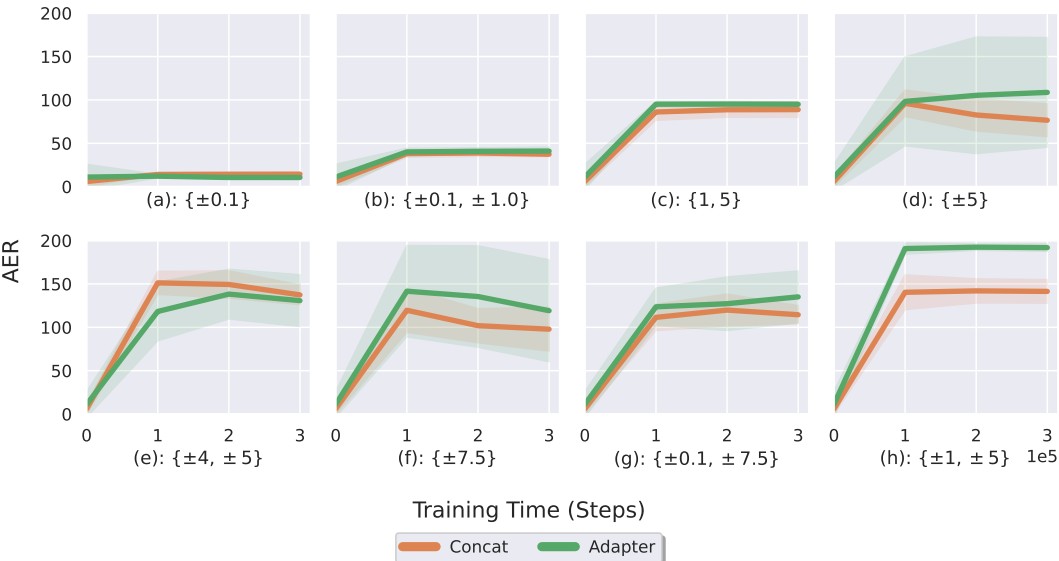

Figure 28: Here we examine the performance when training on different sets of contexts. The text beneath each plot indicates the training context set, with $\pm c$ indicating that both $c$ and $-c$ are in the context set. For instance, in (a), the models were trained on a context set of $\mathcal{C}_{train} = \{-0.1, 0.1\}$. Mean performance is shown, with standard deviation shaded.

### G.3.1 Overfitting

Another potential issue we may encounter is overfitting, which can occur when the training range is not diverse enough. To illustrate this problem, we conduct an experiment in the multidimensional ODE setting using the following set of training contexts:

$$\{(1, 1), (1, -1), (1, 0), (-1, 1), (-1, -1), (-1, 0), (0, 1), (0, -1)\}$$

The results, presented in Fig. 29, demonstrate that the Decision Adapter performs well initially but its generalisation performance suffers as training progresses. Specifically, as shown in Fig. 29b and Fig. 29c, our model exhibits worse extrapolation performance due to overfitting on the narrow training contexts.

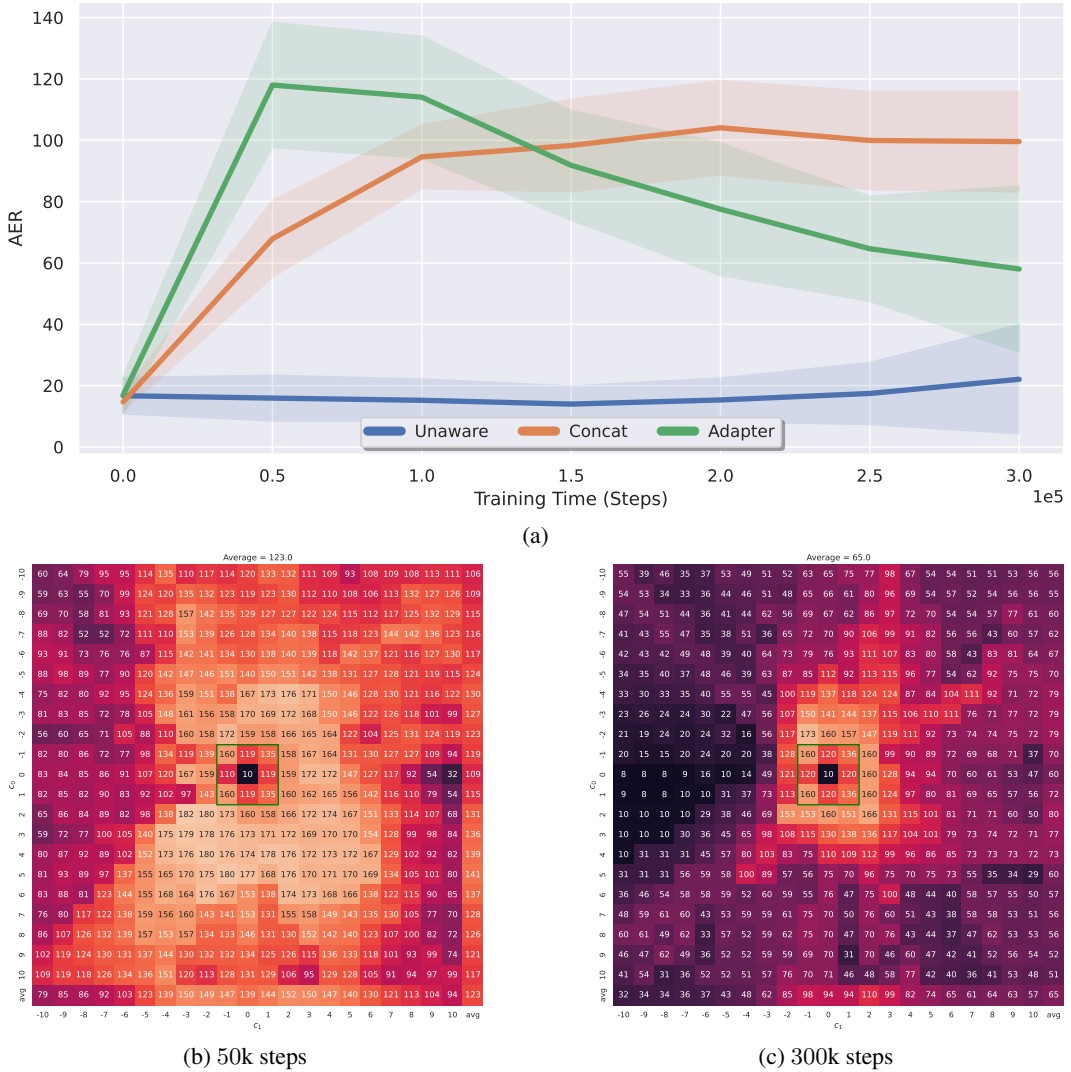

(a)

(b) 50k steps

(c) 300k steps

Figure 29: Showing (a) the overall performance over time on the entire evaluation range. In (b) and (c) we show the performance of the Adapter at two points in training. In (a), we plot mean performance and shade the standard deviation; for (b) and (c), we show only the mean over the seeds.

