# OpenReview forum: "Dynamics Generalisation in Reinforcement Learning via Adaptive Context-Aware Policies"
_NeurIPS.cc/2023/Conference — NeurIPS 2023 poster_

### Official Review · Reviewer_rS79 · 2023-07-04

**Soundness:** 4 excellent
**Presentation:** 3 good
**Contribution:** 3 good
**Rating:** 8
**Confidence:** 5

**Summary:**

The paper proposes a novel way of learning how to adapt policies in contextual RL. The work proposes to use "adapter" modules whose parameters are determined by a hypernetwork that solely focuses on the context variable.

**Strengths:**

The paper is generally easy to follow and well written. Illustrations are well placed and utilized to communicate the content of the paper and to emphasize the concepts that are being discussed.
The work also addresses an important issue for RL, namely generalization, by combining existing ideas in clever ways to tackle this issue. I particularly commend the authors for the discussion of the limitations and the experiments that are conducted to explore the limitations. Overall, the presented method "adapter" seems well suited to be used for training more general RL agents. As such I believe the work is of high significance for the RL community and should inspire future works.

Overall I think this paper is a clear fit for the NeurIPS (RL) community

**Weaknesses:**

The presentation could be improved a bit. For example:
* The title does not seem to fully fit with the work, or rather, it is overly general and gives the impression that a novel cRL formulation will be presented.
* Similarly, the abstract only spends ~1/3 of the space discussing the "Decision Adapter" method
* The wording in line 45 makes it seem like context could not be observed by a sensor. However, the robot mass example would be a sensable context.
* The restriction on only changing dynamics and not changing rewards seems not really needed.
* In line 115, Procgen is cited for examples where context unaware policies can work well, but I'd argue that the context is embedded in the visual state representation
* Theorem 4.5 can only be understood when reading the Appendix. This is the case due to $\mathcal{C}\_{far}$ and $\mathcal{C}_{close}$ never getting defined in Section 4 but only in the appendix.
* Similarly, its easy to overlook why the inequalities in Definition 4.3 make sense, since it depends on the reward definition.
* The theoretical results from Section 4 seem a bit arbitrary since they heavily depend on the construction of the example. It might make sense to move this part to the appendix and move some more of the empirical evaluation to the main paper.
* The AER metric is introduced but "AER" does never show up again in the paper. Is it maybe labeled as "Mean Eval. Reward" in the plots?
* It is not clear why AER can not be defined over a context set that is distinct from the training set
* The citation to Benjamins is by now a TMLR publication



There is one more niche paper using cRL for learning dynamic configuration of AI planning algorithms that should be of interest to you:
* Biedenkapp et al. (2022) https://rlplab.com/papers/biedenkapp-et-al-icaps2022wsprl.pdf (follow up to Speck et al. 2021 https://ojs.aaai.org/index.php/ICAPS/article/view/16008)

This work discusses how to treat state and context information separate and also discusses different architectural choices that one can make in this regard. I don't think this would form a reasonable baseline to compare against, however it feels like an appropriate reference. In particular, it feels like a good reference since the work considers 300+ noisy context features and how to learn with them, compared to the relatively few context features considered in the present work.


While I listed many bullet points for potential improvement, I am very happy with the state of the paper and I think it should be accepted. I am even willing to increase my score if code will be released.

**Questions:**

* Will the code be made available?
* Why is the pre-training phase (as done in NLP) not used for adapter? I would imagine that the hypernetwork would benefit from that and only need to "fine-tune" the generated weights to work well with the observed reward signals.
* With many references to the work of Benjamins et al. 2022, why are not more environments from the proposed CARL benchmark used?
* Intuitively it makes more sense to me to describe the Theoretical foundations in Section 4, using the optimality gap as defined by Benjamins et al. (2022). What is the advantage of using an Optimality ratio instead?
* Are the same context distributions used in the cartpole example as done by Benjamins et al. (2022)?

**Limitations:**

Limitations are discussed thoroughly and experiments are provided that support the discussion on limitations.

---

> ### Author Rebuttal · Authors · 2023-08-09
>
> We appreciate the reviewer's kind words, their thorough review, concrete suggestions for improvement, and endorsement of our in-depth limitations section.
>
> We thank the reviewer for their particular suggestions for improvement, and we generally agree with each point raised. We will rectify each of these points in the revised paper.
>
> > I am even willing to increase my score if code will be released.
>
> We will publicly release our code once the anonymity period is over. In addition, we have provided a link for the AC to a preliminary (anonymised) version of the code.
>
> Please see below for more thorough responses:
>
> - P1-2: We will revise the title and abstract to better reflect the paper's content.
>
> - P3: We did not intend for line 45 to convey this particular meaning and we will clarify it in the revised paper. Our intent was to highlight the qualitative difference in timescale with which state and context change.
>
> - P4: Regarding the restriction on only changing dynamics and not rewards, for clarity we chose to be explicit about defining the restricted problem. However, we do agree that the applicability of our model may be wider. We will mention this in our conclusion.
>
> - P5: We thank the reviewer for noticing the potentially incongruous citation to Procgen. We rather meant to focus on the idea that no explicit context variable was incorporated into the agent, and yet it learned a policy that could generalise well. However, we agree that the context in many Procgen games is embedded in the state. We will remove the mention of Procgen here and discuss it in a more appropriate location.
>
> - P6-7: We will provide the definitions in the main paper. We thank the reviewer for pointing this out.
>
> - P8: We believe formally illustrating example cases where (a) an unaware model fails and (b) where context is not necessary is important for our work. We have also aimed to make the construction of the setup as general as possible. Thus, we believe this represents a broader class of problems. However, we agree that section 4 could safely be shortened, with some of it being moved to the appendix and we will do so. Please see the general comment for more on this point.
>
> - P9: We apologise for the oversight of not being consistent with using the name AER in our plots. The reviewer is correct that "Mean Eval. Reward" is the same as AER. We will rectify this inconsistency in the revised paper.
>
> - P10: It would be possible to define AER in this way, however, the results are very similar (see Figure 4 in the attached PDF). We opted for averaging over the full context range as it provides a better metric of the general performance of the agent, however, we can include the figures for the revised version.
>
> - Regarding Benjamins et al.: While the paper was accepted to TLMR after we submitted our paper NeurIPS, their revised paper is substantially different from the paper originally uploaded to Arxiv. In particular, their revised version no longer introduces or discusses cGate. We will clarify this in our revised paper, and cite both the new TMLR paper (when discussing the work on benchmarks and context in general) and the old Arxiv one that introduced cGate (only when explicitly mentioning cGate).
>
> - Finally, thank you for listing the work of Biedenkapp et al. (2022). We believe the work is indeed relevant and will discuss it in our revised paper.
>
> We next consider the reviewer's questions:
>
>
> - Q2: Our focus is on zero-shot rather than few-shot generalisation (where fine-tuning would be very relevant), so pre-training was not a primary consideration. However, if we were to explore pre-training and fine-tuning, some challenges arise that could be considered by future work: pre-training the entire model raises questions about context selection for pre-training vs fine-tuning, while pre-training only the main trunk limits us to a single context during pre-training.
>
> - Q3: We focus on three environments, ODE, CartPole and Ant. The latter two are in the CARL benchmark, although we used a different implementation. We believe these domains represent a broad range of environments and constitute a representative selection. In particular, the ODE is particularly general and CartPole is a relatively simple classic control environment that is widely used in the literature. Finally, Ant is a more complex and high-dimensional continuous control task and makes the applicability of our work to robotics explicit. We note that there is a trade-off between the breadth of experimentation and depth of results, given a conference length page limit. We hope that we have navigated this trade-off appropriately by deeply exploring the representative and general domains. We will aim to bring more of our additional experiments into the final version of our paper - as the reviewer recommends - which we hope will further justify our experimental strategy.
>
> - Q4: We chose to use the optimality ratio because the optimality gap depends heavily on the reward scale of the environment. For instance, a ratio of 0.1 signifies 10% of the optimal reward. In contrast, the optimality gap can yield varied results, such as 900 (e.g. 1000 - 100), 9, or 0.9, for the same scenario.
>
> - Q5: Our approach differs from Benjamins et al. (2022) [The TMLR version] as we use broader distributions. For example, regarding pole length, we trained on discrete values of 1, 4, and 6, while testing on a range of 0.1-10. In comparison, they tested a narrower range of 0.048-1.19. Our wider range emphasises a clearer distinction between interpolation and extrapolation performance, which we think is an important one to make (as reviewer 8dZQ also notes).
>
> In closing, we again thank the reviewer for their thoughtful comments and overall endorsement of our work. We reaffirm our commitment to publicly releasing our code. We are also committed to actively engaging during the discussion period, and will gladly work to address any further concerns or questions the reviewer may have.

---

> > ### Comment · Reviewer_rS79 · 2023-08-11
> > **Rebuttal Response (Increased Score)**
> >
> > Thank you for the thorough response! My concerns have been sufficiently addressed and I appreciate the new figures that the authors made available. I agree with the authors that they struck a good balance between thorough breadth and depth of experimentation and am not asking or further experiments on CARL environments. I was simply curious why exactly those environments were chosen.
> >
> > I had a look at the code and, as stated in the original review, *I have increased my score*.
> > I am very happy with the paper and am of the opinion it should be accepted. I believe it is a valuable contribution to NeurIPS.
> >
> > I do have additional questions after reading the other reviews. In particular, my questions are along the lines of Reviewer sPQr's Question 5. How do you expect the proposed method would handle the environments of Biedenkapp et al. (2022) (also in comparison to cGate)? Do you believe a hypernetwork would make it easier to deal with such noisy/partially incomplete features? Lastly, would it be a problem for the hypernetwork to have vastly more context features than state features?

---

> > > ### Author Response · Authors · 2023-08-11
> > > **Response to Reviewer's Additional Questions**
> > >
> > > We thank the reviewer for their kind comments, and time spent considering our work.
> > >
> > > Regarding the questions:
> > > > How do you expect the proposed method would handle the environments of Biedenkapp et al. (2022) (also in comparison to cGate)?
> > >
> > > That work is a particularly interesting application of RL. We believe there is no fundamental reason why our approach would not work in this case. Indeed, since some of the features are noisy or not fully informative, our distractor results suggest that we would perform better than either a concatenation-based approach or cGate. That being said, in our we focused our efforts on the continuous-control setting, which is distinct from the discrete policy selection discussed in Biedenkapp et al. (2022) and its related works. Investigating this more in-depth in future work would therefore be promising.
> > >
> > > > Do you believe a hypernetwork would make it easier to deal with such noisy/partially incomplete features?
> > >
> > > As mentioned, we believe that the hypernetwork is particularly well-suited to dealing with these cases where the context variables are potentially distracting. However, as noted in our limitations section (particularly in Appendix G.1), if the context carries more noise than signal, the adapter model's performance suffers. Therefore, while the adapter is quite robust to distractor dimensions, future work can investigate ways to improve its robustness to very noisy contexts.
> > > In particular, one promising avenue would be to more deeply investigate the benefits and potential drawbacks of training in the presence of context noise. This could be expanded to consider a curriculum of noise, starting with a small amount of noise and progressively increasing this as the model learns. In addition, training on a broader set of contexts and associated dynamics (e.g., via domain randomisation approaches) could also improve robustness to noisy contexts and would merit study as well.
> > >
> > > Then, regarding incomplete features, where it may be that the context does not fully describe the environment, or where we cannot perfectly distinguish between separate settings using only the context information: This case is effectively on a continuum between two extremes: (a) A completely context-unaware setting, where we have different dynamics but no way to distinguish between them; and (b) the case we primarily consider in our work, where context does provide perfect information.
> > >
> > > This perspective suggests that the hypernetwork-based adapter module may be promising, as it processes the state features in a similar way to an unaware model, but modulates its behaviour based on the context. Therefore, ideally, if the context provided no information, it would reduce to an unaware model (by effectively implementing an identity function in the predicted adapter weights). Our intuition is that in cases where the context is more useful, but not perfect, the hypernetwork can still partially modulate the behaviour of the agent to take this information into account. Future work could consider this in more depth; one way would be to have a large number of context features (that are all relevant and collectively provide perfect information), and investigate the effect of systematically hiding more and more of these from the agent.
> > >
> > > > Lastly, would it be a problem for the hypernetwork to have vastly more context features than state features?
> > >
> > > This is one aspect that makes the work of Biedenkapp et al. (2022) quite interesting; there are only 5 dynamic features (i.e., state) per open list but 305 instance features (i.e., context). While we have not explicitly considered a case like this, we believe the hypernetwork will perform well for the following reasons:
> > > 1. Empirically, in the 1D ODE domain, our distractor results (which are very similar to the other two domains) considered up to 100 distractor dimensions; totalling 1 informative context dimension, 1 state dimension and 100 uninformative context dimensions (resulting in a similar ratio to Biedenkapp et al. (2022)). The adapter performed admirably and could successfully solve the task, and generalise to unseen instances.
> > > 2. Due to the adapter's architecture, where it explicitly separates the state and context features, we believe that it would better deal with cases where there are significantly more context features. This is because the state-based features are not immediately combined with the context-based ones (potentially running the risk of the model focusing on the many context features as opposed to the few state features).
> > >
> > > One potential problem in this case is overfitting, especially if we only have access to a small set of environments, and therefore very sparse coverage of the (large) context space. However, this problem is not unique to hypernetworks and would likely affect many different models, including concat and cGate.
> > >
> > > We believe these points merit discussion in the revised paper and we will make an effort to include them.

---

### Official Review · Reviewer_DXBg · 2023-07-05

**Soundness:** 2 fair
**Presentation:** 3 good
**Contribution:** 2 fair
**Rating:** 6
**Confidence:** 4

**Summary:**

This paper addresses the problem of learning policies that can generalize to multiple contexts. The problem is formulated as a contextual MDP (CMDP). The paper starts by analyzing analytically the performance of policies that are not aware of the context in two different scenarios of a particular instance of CMDP. After that, a network architecture (decision adapter), which incorporates contextual information through a hypernetwork, is introduced. The architecture is empirically evaluated in three environments and its performance is compared against 4 baselines.

**Strengths:**

The paper tackles an important problem in RL. The idea of using a hypernetwork to incorporate contextual information is sound and the empirical results are promising. The paper is very clear and easy to follow.

**Weaknesses:**

I believe that the paper's novelty is somewhat limited. While I found the solution of using a hypernetwork quite neat, I was slightly disappointed to discover that a similar architecture called cGate had already been proposed to address the same problem. The cGate architecture is mentioned for the first time in Section 5 and is not introduced as a related work.

The authors argue that the adapter architecture is a more powerful generalization of cGate as it can also capture non-linear relationships between states and contexts. This is confirmed by the empirical results obtained in the ODE domain. However, the results of cGate on the other two environments (CartPole and Mujoco Ant) are not reported. Moreover, the ODE domain seems to be specially designed to make cGate fail, as the dynamics are given by an nth-degree polynomial (equation Line 229).

Finally, I found the example provided in Section 4 to be very helpful for understanding. However, in my view, framing the analysis o as a theorem is not appropriate in this case, since the applicability of the results seems limited to this specific instance of the problem. Typically, a theorem should offer broader insights into the overarching problem at hand.

**Questions:**

1. Could the authors clarify why the results of cGate in CartPole and Mujoco Ant are not provided?

2. The paper attributes cGate to [13]. However, I did not find any reference to cGate in [13]. Could the authors clarify this?

3. Line 193 says *"For this and the next section, we consider only the Concat and Unaware models as baslines"*. However, the results for the unaware policy are not reported in Figures 4 and 5.

**Limitations:**

The limitations of the approach are discussed extensively in Section 8 and Appendix G.

---

> ### Author Rebuttal · Authors · 2023-08-09
>
> We appreciate the reviewer's acknowledgement of the importance of this problem in RL, and appreciate their saying that our paper is clear and easy to follow.
>
> Regarding the weaknesses raised by the reviewer:
>
> Firstly, we agree that cGate should have been discussed more thoroughly in the related work and we will rectify that in the revised paper. Secondly, cGate (which to date has only appeared in an Arxiv submission - however we are aware that NeurIPS policy is to consider this in prior work) is indeed a special case of our model. Although, it was not designed or introduced as a hypernetwork and conceptually our work is the first to take this perspective of cGate. Thus, we have drawn the comparison with cGate in an effort to accurately ground our work in the literature - but only with our change in perspective on cGate is it clear that an adapter layer with hypernetwork is an appropriate more general model. However, we emphasise that this requires an understanding of the mathematics driving cGate and was not a conceptually trivial point. Thirdly, we have run more experiments to ascertain the benefit of our model over cGate and indeed see that our model provides an empirical advancement whether or not the context contains distractor variables. As to the point that the ODE setting is adversarial towards cGate due to the use of an n-order polynomial. We would like to emphasise that the context provides the coefficient of this equation and no power is applied to these coefficients. In other words, the context features being input to cGate are appropriate for learning the task (unlike if some context $c_i$ features was used when $c_i^2$ is in the equation, in which case this would be needlessly adversarial). Indeed, cGate is able to learn on the trained contexts and the appropriate function is in the hypothesis space of the cGate model (see Fig. 3 in the main paper, cGate vastly outperforms unaware, but does perform worse than concat. In addition, Fig. 10 in the appendix shows that cGate is able to perform optimally on the training contexts, it suffers primarily in extrapolation).
> Finally, to say that ``cGate had already been proposed to address the same problem'' is not quite accurate. This is due to the second point - it was not introduced as a hypernetwork, and moreover, cGate was not used to adapt to distractor context variables. It could not have because, as our new experiments show, it offers no benefit there over the concat model. We thank the reviewer for pointing out the need for us to be clearer about the origins of cGate so that readers are better able to determine the contributions of our work. We hope that we have clarified the relationship between cGate and our hypernetwork, as the hypernetwork is more than just cGate with a nonlinearity.
>
> Third, we aim to have Theorem 4.5 as a particular example (and formal showcase) of certain cases where (a) an unaware model fails and context is necessary and (b) an unaware model succeeds and context is not necessary. However, based on this reviewer's and the other reviewers' comments, we will sightly shorten our theoretical section to make room for additional empirical results (see the general comment above). Regarding the scope of Theorem 4.5, we agree with the reviewer that it is not as broad as a theorem generally should be, and will instead refer to it as Proposition 4.5
>
>
> Regarding the reviewer's questions
>
> 1. We only provided one other context-aware baseline, the concatenate model as a representative baseline of this class of methods. We believe this model is representative, as it is very widely used, and performed similarly to, or outperformed cGate in our prior experiments. However, in the attached figure PDF, we have the results for cGate on CartPole and Ant (Figure 2). We note that cGate fails in a similar fashion to the concatenation model in CartPole. In Ant, cGate also exhibits reduced performance when the number of distractor variables increases. In addition to this, cGate performs worse than the Adapter and Concat models when no distractor variables are present in Ant (at the end of training, cGate has a mean reward of around 2800, whereas the adapter/concat models have rewards of around 4000). We will add these results for CartPole and Ant to the revised paper as we think they further support the use of our model.
>
>
> 2. Regarding reference [13], we noticed that the authors of [13] uploaded a new version of their paper to Arxiv, on the 2nd of June 2023 (after we submitted our paper). Their updated paper is significantly different to the one we cite (which is ``v1'' on ArXiv) and removes mention of cGate entirely. This is a slightly unusual case to be in as this would depict cGate as no-longer being prior work (at least based on the latest state of ArXiv). However, to be fair to the authors of [13] we will cite v1 and v2 of this paper separately. Hopefully then readers who would like to understand cGate can find the earlier version but for all other points where we cite [13] the  most up-to-date version will be used. As mentioned above we will still add [13] v1 to our prior works section, however, it seems prudent to avoid grounding our work too fully in an out-dated version of this paper.
>
> 3. Thank you for pointing out that issue. We apologise for the oversight. The focus of these experiments was primarily on how the concatenate model differs from the adapter, in its robustness to distractor context variables. Therefore, the unaware model is not a valid comparison in this case and we will remove the mention of the unaware model in this line.

---

> > ### Comment · Reviewer_DXBg · 2023-08-13
> > **Response to authors**
> >
> > I would like to thank the authors for addressing my concerns.
> >
> > After reading the rebuttal I still have a few questions:
> >
> > - *Authors: "cGate is able to learn on the trained contexts and the appropriate function is in the hypothesis space of the cGate model".*
> >
> >    **Response:** If the underlying function is in cGate's hypothesis class, how do you explain the poor performance of cGate on ODE?
> >
> > - *Authors: "We hope that we have clarified the relationship between cGate and our hypernetwork, as the hypernetwork is more than just
> >   cGate with a nonlinearity."*
> >
> >    **Response:** Line 196 in the paper says: *"Finally, we note that our adapter architecture here is a more powerful generalisation of another
> >    network architecture introduced by prior work, cGate [13]. Intuitively, our model uses a nonlinear neural network to process the state
> >    based on the context, whereas cGate uses only a single, linear, elementwise product."*
> >
> >    Could the authors clarify in what way the use of a hypernetwork is beneficial other than by being nonlinear?
> >
> > I kindly ask the authors to be concise in their response. The rebuttal was quite difficult to parse.

---

> > > ### Author Response · Authors · 2023-08-14
> > >
> > > We thank the reviewer for their time spent considering our work and their engagement in the review process.
> > > While endeavouring to respect the reviewer's request for concision, we acknowledge that there are several helpful perspectives to take to answer these questions. We will attempt to be brief and urge the reviewer to continue this discussion if in our brevity something remains unclear.
> > >
> > > > If the underlying function is in cGate's hypothesis class, how do you explain the poor performance of cGate on ODE?
> > >
> > > Figure 3 shows the overall performance of the agent across the entire context range. Importantly, this contains contexts that the model did not train on, as the goal of our work is to improve context generalisation. Thus, the poor performance of cGate in Figure 3 (left) is due to its failure to generalise and *not* because it cannot fit the seen contexts. To support this and explain the poor performance we next direct the reviewer's attention to Figure 10 in the appendix. This shows the same information as Figure 3 (left), but separately considers the models' performance on training, interpolation, and extrapolation contexts, respectively. Here, we note that cGate performs perfectly on the contexts it trains on - confirming that the ODE domain is within the hypothesis space of cGate - but fails to extrapolate.
> > > Interpreting these results and the improvement afforded by a hypernetwork, cGate's inductive bias may be responsible for this poor generalisation.
> > >
> > >
> > > > Could the authors clarify in what way the use of a hypernetwork is beneficial other than by being nonlinear?
> > >
> > >
> > > We emphasise that cGate is  **not** a hypernetwork. Rather it is an implementable architecture **by** a hypernetwork. This is an important but subtle point which we contribute to the literature. Thus, while also allowing for the addition of non-linearity, hypernetworks are also likely to have different inductive biases and training dynamics compared to cGate, which could also be the source of the improved generalisation. Put simply, while our model could implement the same function as cGate, it does not appear to.
> > >
> > > Furthermore:
> > > - As we have empirically demonstrated in this work, our model displays superior generalisation performance and improved robustness to distractor variables compared to cGate. This alone indicates that the model has meaningfully different behaviour which cannot be merely attributed to the presence of non-linearities (not without extensive future works).
> > >
> > > - Indeed, other fields such as NLP [1,2,3] and Computer Vision [4] have also identified and begun to explore the benefits of hypernetworks and adapter layers, which further grounds our approach. Yet much work remains in understanding neural network generalisation broadly, let alone the incorporation of something as dynamic as a hypernetwork. Further to this, our more general architecture could easily be modified to incorporate more sophisticated architectural designs with different inductive biases, such as convolutions.
> > >
> > >
> > > We hope this has clarified these aspects, and we are committed to engaging further in this discussion if the reviewer has any additional questions.
> > >
> > > References
> > >
> > > [1] Neil Houlsby et al. Parameter-efficient transfer learning for NLP. ICML, 2019.
> > >
> > > [2] Rabeeh Karimi Mahabadi et al. Parameter-efficient multi-task fine-tuning for transformers via shared hypernetworks. ACL/IJCNLP, 2021.
> > >
> > > [3] Haoxiang Shi, Rongsheng Zhang, Jiaan Wang, Cen Wang, Yinhe Zheng, and Tetsuya Sakai. Layerconnect: Hypernetwork-assisted inter-layer connector to enhance parameter efficiency. COLING, 2022.
> > >
> > > [4] Sylvestre-Alvise Rebuffi, Hakan Bilen, and Andrea Vedaldi. Learning multiple visual domains with residual adapters. NeurIPS, 2017

---

### Official Review · Reviewer_sPQr · 2023-07-06

**Soundness:** 3 good
**Presentation:** 3 good
**Contribution:** 2 fair
**Rating:** 6
**Confidence:** 4

**Summary:**

The paper studies how to better incorporate context (such as MDP-specific parameters) in context-conditioned robot policies. Traditionally, this is done simply by concatenating the context vector to the state vector and processing both inputs simultaneously with the policy network. This work proposes a context-conditioned hypernetwork to output the weights for an adapter module, which adds a residual value to the policy’s intermediate representation. Along with other proof-of-concept tasks, the paper evaluates on cartpole (varying pole length) and mujoco ant (varying mass) and finds that the proposed method is more robust to distractor dimensions but does not necessarily perform better than naive concatenation methods.

**Strengths:**

The authors study a problem that is important in multitask learning/generalization such as with respect to language/goal-conditioned policies.

The paper is well-written and proves some interesting theoretical results. Empirically, experiments test generalization in a meaningful way. For instance, the evaluated mass ranges are not within the training ranges in the ant environment, which is a challenging problem in robotics.

**Weaknesses:**

(1) Section 4 talks about performance guarantees for a context-unware policy. Why is this important to the rest of the paper, which is about how to best incorporate context into a policy for dynamics generalization? Furthermore, I don’t see the link between Lines 154-163 (talking about how context-dependent rewards are equivalent to context-dependent dynamics in a context of matrix rotation example) to the rest of the section or the rest of the paper.

(2) The abstract promises to “establish conditions for when it is necessary to incorporate context,” but when reading the text, I did not see any section of the paper that brought this out explicitly.

(3) The work would really benefit from architectural ablations, especially when the abstract said the paper would “examine how to best leverage this external context information to improve generalization.” Hypernetworks are not a common architectural component in context-conditioning, so were alternatives not involving hypernetworks considered? For instance, it would be good to compare with the following architectural results:
(i) The proposed approach, with no hypernetwork, where the adapter module (a simple MLP) only takes the context as input.
(ii) Something like cGate, but with an elementwise product with the context embeddings at every layer, instead of just at one layer.

(4) Figures 4 and 5, which contain the most difficult domains examined by the paper (ant and cartpole), show that in the 0 distractor dimension case, naive concatenation performs roughly the same as the proposed method (adapter). This means that in this case, naive concatenation is preferable due to its simplicity and training speed relative to the proposed adapter. It is only in the differential equations domain, a non-control task, where adapter clearly outperforms naive concatenation.

**Questions:**

(1) How might this framework be improved to incorporate multiple contexts at the same time? For instance, both a goal image and language instruction for a given task. If it is known that some context modality is more beneficial on average to inducing strong task performance than some other context modality, how might we handle that with adapters and hypernetworks?

(2) I don’t understand how Theorem 4.5 relates to the rest of the paper, or even how it guided the experimentation toward the proposed adapter + hypernetwork architecture.

(3) How would the proposed method perform in image-based domains where the main policy network is a CNN or transformer? It seems like the main results are working with an MLP operating on robot state. My main concerns are that this work may be useful primarily for simpler, older architectures and might not prove as helpful for transformer-based robotics policies that are moving toward cross-attentional context conditioning (such as VIMA [Jiang et al 2023]).

(4) Perhaps include a citation of FiLM (Perez et al 2017), since it is a widely used context conditioning method and does a sort of context-conditioned “adapter” with a residual connection as well, which is similar to the proposed method but does not have a hypernetwork. A comparison would be helpful too.

(5) Most of the crux of the experimental section hinges on the argument that the adapter is a better context conditioning architecture because it is more robust to distractor context variables. However, the distractor variables were set to some pretty contrived values--all to 1 during training and 0 during testing. How were these values chosen? Why not just gaussian noise during both training and testing, which seems more likely for a model to encounter, in which case concat will probably be as good as an adapter at filtering out the distractor dimensions?

**Limitations:**

Authors were upfront of the limitations of the adapter to noisy contexts and its tendency to overfit when trained on narrow context ranges and had a very detailed limitations section.

---

> ### Author Rebuttal · Authors · 2023-08-09
>
> We thank the reviewer for their detailed review. We hope that we can address the reviewer's concerns and questions. In particular, we would like to direct the reviewer's attention to the additional PDF with the requested experiments of the additional baselines, which we elaborate upon below.
>
> Regarding the noted weaknesses:
>
> - 1. and 2. and Q2. Please see the general response for more elaboration. In essence, we believe Theorem 4.5 motivates the need for context-aware policies. However, the theorem, and its link to the rest of the work should be explained more clearly. We thank the reviewer for raising these points and helping us tie all results of this work together better.
>
> - 3. We agree that a more in-depth look into architectural ablations is important and we thank the reviewer for giving concrete suggestions. We performed these experiments on the ODE, with the results in the attached PDF (Figure 1). We considered:
>    - `AdapterNoHnet`: This is our adapter module without a hypernetwork. The adapter's architecture and location are the same, but it is now a single MLP that takes in the concatenation of the state-based features and the raw context (as the model must be context-conditioned, and must also process the state-based features).
>    - `cGateEveryLayer`: This is exactly what the reviewer suggested, the same elementwise operation as cGate, happening at every hidden layer instead of just one.
> Both of these ablations perform worse than our adapter and cGateEveryLayer does not outperform the standard cGate.
> We have also run several additional ablations investigating certain aspects of our adapter. We considered (1) the adapter module's architecture, (2) the hypernetwork's architecture and (3) the location of the adapter. For space reasons, ablation (2) is shown in Figure 3 in the attached PDF, and we will add the others to our paper. Overall, these experiments show that the adapter is generally robust to these hyperparameters, with most values performing comparably. We believe these additional results further strengthen our work.
>
>
> - 4. We believe the concat model performing comparably to the adapter when no distractor variables are present demonstrates the source of the concat method's vulnerability. Specifically, treating context as a state feature results in overfitting. The hypernetwork, however, is more robust and less prone to generate weights for the policy network that are overfit. Thus, in a setting where the context is completely reliable, and the notion of overfitting is not applicable, then concat and the hypernetwork are indeed comparable (in some domains). This is an important finding for a work which aims to identify *when and how* to incorporate context in RL. In the case of completely reliable context, then, it is unnecessary to use a more sophisticated model. However, in light of the other literature in context-sensitive RL, this is unlikely to be true in many cases - particularly those in which context is inferred or we are unsure about which context variables to incorporate. We thank the reviewer for drawing our attention to this point, and we see that this should be explained more fully in the paper as it is an important one. In sum, *when* your context may include distractors then *a hypernetwork is the appropriate model*
>
>
>
> We next consider the reviewer's questions:
>
> - 1. and 3. One way to incorporate multiple context modalities would be to have one or more models in front of the hypernetwork that serve to transform the different modalities into embeddings (i.e., continuous vectors) which are subsequently concatenated. We would also argue that by using a hypernetwork it is possible to then learn, based on the task, which context information is more valuable. Indeed, if most of the useful contextual information can be found in one modality then this would make the other modality akin to a distractor context. Based on our results, this would suggest that the hypernetwork would appropriately handle this situation. Similarly, we believe there are no reasons that our approach would fail in image-based domains. One approach would be to have a CNN model process the image observation into a flattened vector, which is then passed to an MLP that outputs the action. We could add our hypernetwork to this MLP. We believe that we have taken a step towards robust context-sensitive RL agents in this work and are excited to see that the reviewer has immediate directions of future work in mind. Unfortunately, multi-modal learning (let alone multi-modal RL) is a challenging and important problem in its own right. Thus, it would seem to fall outside of the scope of this work.
>
>
>
> - 4. We apologise for the oversight of not citing FiLM, we will discuss it in the revised paper. We note that cGate is functionally very similar to FiLM, as both perform a single elementwise multiplication between the main (state) and external (context) features. Therefore, FiLM, like cGate, is still linear, in contrast to our model. Finally, we chose to focus on cGate as it was more recently introduced than FiLM, and was specifically introduced to address the problem of generalising to new contexts (as opposed to FiLM which is quite general).
>
> - 5. Regarding the distractor experiments: here, we aim to study the effect of having distractor variables in the context (that do not affect the dynamics), but that have different values between training and testing. To this end we chose $0$ and $1$, to indicate that during training, we had some context variables at particular (but fixed) values $X$; during testing it changed to other (still fixed, but different) values $Y$.
> Thus, to the point of ``Why not just gaussian noise during both training and testing'' - there is no guarantee that the sampled context values will be meaningfully different to demonstrate a true test set of contexts. Thus we required a more explicit approach.  We will clarify and expand upon this explanation in the revised paper.

---

> > ### Comment · Reviewer_sPQr · 2023-08-20
> > **Response to Rebuttal**
> >
> > Thanks very much for your effort in your response. I find your response to weakness 3 and 4 to be great, and thanks for running those additional experiments. I also find your answers to questions 1 and 3 to be reasonable approaches.
> >
> > My main remaining concerns are about linking the theoretical part to the experimental part of your paper in a much better way. It's a bit clearer to me after reading your rebuttals, but for example, lines 153-162 are still really hard for me to connect to the thesis of the paper even after I have read it several times again after your rebuttal. I appreciate that you are willing to improve this in the revision.
> >
> > For the distractor experiments, I still think it's not great to be using 0's and 1's for distractor variable values. I take your point about wanting the distractors to have different values between training and testing, but such well-structured binary values for distractor variables may cause weird things to happen in continuous-valued neural nets, as they basically render parts of the network weights useless when multiplying by the weight matrix $W$ in $XW+b$.
> >
> > 1. Could you look into perhaps sampling from $N(0, \sigma)$ during training and $N(1, \sigma)$ during testing for the distractor variable values, where $\sigma = 0.2$ for example? That would create two (practically) separate distributions for the distractors during training and testing.
> > 2. Do the advantages of the adapter hold over the concat method if there are 20 distractor variables and each of them is sampled from $N(0, \sigma)$ during both training and testing? I know I am pressing on this--it's just in my mind, I interpret 'distractor variable' as something that is irrelevant to the task context, not necessarily something that must be different during training vs testing AND is also irrelevant to task context AND has fixed values.
> >
> > I think having the answers to these two questions would be important as it would show to me clearly when a hypernetwork is important for context vs simple concat. Thank you very much for engaging in a good discussion.

---

> > > ### Author Response · Authors · 2023-08-20
> > >
> > > We thank the reviewer for their engagement in the discussion process, and praising our responses to several of the weaknesses and questions raised in the original review.
> > >
> > > Regarding the theoretical linking, the core idea behind those lines is the following:
> > > 1. The theoretical section considers a case where the dynamics of the environment is consistent (regardless of context) but the reward changes depending on the context.
> > > 2. The experimental section considers the dynamics being affected by the context, and reward remaining consistent across context (noted in lines 79-80).
> > > 3. Therefore, we need a way to link the theoretical section (consistent dynamics and varying reward) to the experimental section (varying dynamics and consistent reward). Lines 154-163 formally make this link, noting that there is an equivalence between (1) and (2) and making it clear that the theoretical results apply to the experimental setup considered.
> > >
> > > As mentioned in the general rebuttal, we will move this to the appendix for readers that still seek such rigour, but we agree that it should not be in the main paper. We will also make it clear why this discussion is relevant, along the three points mentioned above. However, we reiterate that the theoretical section is necessary as it discusses the case for *when* incorporating context is necessary. Please see the general rebuttal for a longer comment on the necessity of this section. However, we hope the elaboration above further addresses the reviewer's concern and makes the link between sections clearer.
> > >
> > > > My main remaining concerns are about linking the theoretical part to the experimental part of your paper in a much better way
> > >
> > > On this point, we would be eager to discuss any suggestion on how to link the sections better before the end of the discussion period. Does the reviewer have any proposed improvements - both compared to our original version as well as to the proposed changes mentioned above and in our original rebuttal?
> > >
> > >
> > > We will gladly run the requested experiments where, instead of being set to 0/1, the distractor variables are sampled from a Gaussian distribution.
> > > However, since the discussion period ends tomorrow, we are unsure if they will be finished before then. Regardless, we can add these experiments to the revised paper.
> > >
> > > To clarify, for these new experiments, we sample new distractor values only once per episode. This is because we require the context to be consistent within a single episode. This strategy is in contrast to sampling new distractor context variables at each step, which would conflict with our original distinction between state and context variables.
> > >
> > > Having noted this, we still believe the distractor experiments we have in the originally submitted paper are useful.
> > > The fact that these variables have different values between training and testing is a good showcase of how robust models are. Furthermore, we believe this choice is very important, since we measure the ability of agents to **generalise** to new settings. It is quite feasible if the agent is in a scenario where the relevant context variables change, that the irrelevant ones may also be different to what it experienced during training.
> > >
> > > Then, regarding having fixed values, we believe this also makes sense in certain cases. For instance, if we consider a robotics task, all training could be done in specific rooms in one particular building where each room has different properties - such as the floor friction. In this case, irrelevant context features (e.g. humidity, air pressure, etc.) would correspond to our distractor variables and would remain fixed during the entirety of training (as training is only done within this one building). Then, if testing is in a completely different area, these irrelevant variables may be different from what was encountered during training. This is increasingly likely when we consider the life cycle of a robot where conditions will change over time. Thus, we still believe our experiments are demonstrative of very important real-world settings and use cases that are not contrived.
> > >
> > >
> > > We hope this further clarifies these points. Unfortunately, we may be unable to get all of the results in one day - however, we will endeavour to give the reviewer as much information on these experiments as possible by the end of the discussion. If the reviewer has any more concerns in the meantime we would be happy to respond further as the discussion period draws to a close.

---

> > > > ### Comment · Reviewer_sPQr · 2023-08-21
> > > > **Re: New distractor experiments**
> > > >
> > > > Thanks again for your prompt response! I know that the timeline is very tight, and I'm sorry that I should have responded to your rebuttal much earlier--I had not expected the implementation time to be more than a few hours for my proposed distractor experiments, but forgot that I do not know what your training time is like. No worries if the training doesn't completely finish in time by Monday; if you can show any preliminary results (like the first x timesteps, if that is usually a signal of how the methods perform relative to each other at convergence), that would be appreciated. (I do want to note for the record that I did propose this experiment at a higher level in my very first review in Question 5:
> > > >
> > > > > However, the distractor variables were set to some pretty contrived values--all to 1 during training and 0 during testing. How were these values chosen? *Why not just gaussian noise during both training and testing, which seems more likely for a model to encounter,* in which case concat will probably be as good as an adapter at filtering out the distractor dimensions?
> > > >
> > > > Re: fixed values, I agree that it does make sense in certain cases with env attributes that you mentioned, like friction. I was specifically dubious about the choice to use 0's and 1's specifically. Point taken that your method is better at generalizing to these 0/1-valued dimensions during testing, but it's unclear if this will be the case with other fixed values or other random-noise values.
> > > >
> > > > Thanks again for doing your best to address my concerns and answer my questions. I just wanted to get this comment out, and in my next comment (which I'll try to finish on 8/21), I can list out a few more concrete points for connecting the theoretical to the experimental part better after thinking a bit more about what you wrote above.

---

> > > > > ### Author Response · Authors · 2023-08-21
> > > > > **Additional Distractor Results**
> > > > >
> > > > > We again thank the reviewer for their engagement in the discussion process. We also thank them for considering concrete improvements to the linking; we will be sure to take these into account when preparing the revised paper.
> > > > >
> > > > > As a follow-up to yesterday's message, we have managed to finish running the first set of experiments, with Gaussian distractor variables. We summarise the results below, but these new results do not change our conclusions; in fact, they strengthen our claims that the adapter is more robust to distractor variables than the concat model. We thank the reviewer for suggesting these additional experiments.
> > > > >
> > > > >
> > > > >
> > > > > In particular:
> > > > > - We ran these experiments on the ODE and CartPole domains, but we can run these on Ant for the revised paper as well.
> > > > > - We trained for 300k steps, but the reward curves started to plateau around 100-200k steps, so the models had sufficient training time.
> > > > > - To retain parity with the results in the main paper (where the distractor during training was $1$ and during testing it was $0$), we set the training context to be $\mathcal{N}(1, \sigma)$ and the testing contexts to be $\mathcal{N}(0, \sigma)$; however, we can run the opposite as well for the revised paper if so desired.
> > > > > - We used $\sigma=0.2$ as requested.
> > > > > - We ran over 8 seeds instead of the usual 16 (due to computational/time constraints), but the error bars are not much larger than those in Figure 4. For the revised paper all experiments will be run over 16 seeds.
> > > > >
> > > > >
> > > > > While we have contacted the AC to determine if we are allowed to share the figures at this point, here we briefly summarise the results:
> > > > > ### ODE
> > > > > - The concat model's plots look similar to Figure 4 (left) in the main paper. In particular:
> > > > >     - The concat model with 0 distractor variables performs comparably to 1 distractor variable, achieving a final AER (the metric defined in section 6.1, the same one used for all of the other plots) of around 140-150.
> > > > >     - The concat model with 20 or 100 distractor variables performs much worse, with around 50 AER at the end of training.
> > > > > - The adapter model's plots look similar to Figure 4 (right) in the main paper. In particular:
> > > > >     - The adapter performs comparably with 0, 1, 20 or 100 distractor variables. The AER at the end of training for 0 or 1 distractors is around 190. The AER at the end for 20 or 100 variables is slightly lower at around 160-170.
> > > > >
> > > > >
> > > > > ### CartPole
> > > > > For CartPole, we also have very similar results. In particular, as we increase the number of distractor variables we have, the concat model performs worse.
> > > > > - At 0  distractors, the concat model has AER=~450
> > > > > - At 1 distractor, it has AER=~360
> > > > > - For 20 distractors, AER=~280
> > > > > - For 100, AER=~220
> > > > >
> > > > >
> > > > > The adapter, however, displays much less of a dropoff. Each setting (i.e., 0, 1, 20 or 100 distractor variables) has a similar AER between 450 and 350.
> > > > >
> > > > > ### Summary
> > > > > Therefore, these additional results further strengthen the claim that adapters are more robust to distractor variables, even if these are sampled from a Gaussian.
> > > > >
> > > > >
> > > > > We hope these results alleviate the reviewer's concerns regarding our distractor variable experiments.

---

> > > > > > ### Author Response · Authors · 2023-08-21
> > > > > > **Additional results where the training and testing distractor distributions are the same**
> > > > > >
> > > > > > Dear Reviewer sPQr,
> > > > > >
> > > > > > We have contacted the AC regarding how best to share the figures for these and the previous new results, since we are not allowed to share links according to the NeurIPS guidelines. We hope to be able to share these soon but for the moment we will report our findings.
> > > > > >
> > > > > > We have run the second set of experiments, in particular, the noise is sampled from $\mathcal{N}(1, \sigma)$ both during training and testing.
> > > > > > This is to be consistent with the experiments we had in the previous message, where training was done on $\mathcal{N}(1, \sigma)$.
> > > > > > We again use $\sigma=0.2$.
> > > > > >
> > > > > > ### Results
> > > > > > The results for both ODE and CartPole in this case are similar. Particularly, for both the concat and adapter models, there is not a significant difference between having 0 distractors and 20 distractors.
> > > > > >
> > > > > >
> > > > > > For completeness, we also train on $\mathcal{N}(0, \sigma)$, as requested.
> > > > > > In this case, for the ODE, we observe a similar result compared to $\mathcal{N}(1, \sigma)$
> > > > > > The adapter's performance with 20 distractor dimensions is similar to its performance without any distractors.
> > > > > > The concat model seems to learn slightly slower when it has 20 distractor variables; its performance lags slightly behind the model without these variables.
> > > > > >
> > > > > > On CartPole, the concat model's performance with 20 distractors again lags a bit behind the results with 0.
> > > > > > For the adapter, as noted in the main paper, an overly large signal-to-noise ratio may negatively impact the adapter's training (see e.g. Figure 12(c) in the appendix). In this particular experiment (CartPole, training and evaluating on $\mathcal{N}(0, \sigma)$), we notice a similar effect, i.e., that the variance is somewhat high for the adapter when training on 20 distractor dimensions.
> > > > > > However, if we consider the generalisation gap, i.e., the difference between training and testing performance (which we are interested in, as our paper focuses on improving generalisation), the conclusions are similar to the ODE results, in that having distractor variables does not negatively impact the performance.
> > > > > >
> > > > > >
> > > > > >
> > > > > > ### Summary
> > > > > > In summary, we find that if the distractor distributions are kept constant between testing and training, both the adapter and concat models perform similarly to having no distractors.
> > > > > >
> > > > > > However, we wish to note that this experiment does not test generalisation performance with respect to distractor variables.
> > > > > > In particular, if the concat model suffers from overfitting (as seems to be the case in all of our distractor variable experiments), the overfitting is not demonstrated when we evaluate on in-distribution data (similarly to supervised learning, overfitting is only visible when evaluating generalisation performance).
> > > > > >
> > > > > > For these reasons, we believe these results could be added to the revised paper's appendix, and discussed to contextualise the distractor results - The adapter is significantly less susceptible to overfitting to irrelevant distractors compared to the concat model (this conclusion from the original paper remains unchanged, and is further strengthened by the results described in the previous message).
> > > > > > However, if these distractors remain the same during training and testing (which may not be particularly likely), it does not matter very much whether the model overfits or not.
> > > > > >
> > > > > >
> > > > > > We again thank the reviewer for their engagement, overall kind words in the responses, and suggestions to improve our paper.

---

> > > > > > > ### Comment · Reviewer_sPQr · 2023-08-22
> > > > > > > **Response to Author's additional Distractor experiments, Updated Score, Theoretical-Empirical Linking, and Further Suggestions**
> > > > > > >
> > > > > > > **Re: New distractor experiments that I requested.**
> > > > > > >
> > > > > > > I really appreciate the authors sprinting to get the gaussian noise distractors experiments in. I am glad that the proposed hypernetwork method is robust to $\mathcal{N}(1, \sigma)$ distractors during training and $\mathcal{N}(0, \sigma)$ during testing and believe this is a stronger and more impressive result than the fixed value setting in the paper.
> > > > > > >
> > > > > > > I have decided to update my score from 4 (weak reject) to 6 (weak accept) because most of the weaknesses and questions I listed in my original review were addressed to my satisfaction.
> > > > > > >
> > > > > > > I also reread portions of the paper and found lines 78-79: “Our goal is to train on a particular set of contexts, such that we can generalise well to another set of evaluation contexts.” It now makes sense to me why the authors really wanted distractor contexts that were different. However, I think that even if the distractor dimensions are of the same distribution but the relevant context dimensions are different between train and test, this still falls under the desired generalization problem setting. Thus, I still think that the train-test-identically-distributed set of distractors that the authors ran in the comment above is relevant for the paper. The authors' finding above, that concat and the proposed hypernetwork method perform roughly the same over increasing distractor dimensions, makes sense to me, and would help readers understand when to use concat and when to use hypernetworks, given hypernetworks are probably more expensive to train.
> > > > > > >
> > > > > > > **Re: Connecting Theoretical and Empirical parts of the paper better**
> > > > > > >
> > > > > > > The abstract of this paper addresses when to use context and how to architecturally do so. The theoretical section talks about when to use context, and the empirical section presents the best architectural method to use context, assuming that one decides to do context-conditioning. There’s a fairly strong separation between the two sections, unless this was intended.
> > > > > > >
> > > > > > > 1. For instance, the key result of the theoretical section was Theorem 4.5, mathematically stating in a toy MDP the performance bounds on how well a context-conditioned policy does compared to an unaware policy. Does anything in the experiments depend on Theorem 4.5? As far as I can tell, no, in part because it is so specific to the toy MDP (as reviewer DXBg has brought up separately).
> > > > > > >
> > > > > > > 2. In the experiments, there is also no attempt to try to quantify the $\alpha^{\pi}(s)$ ratio, which undergirds the theoretical section. This could be one potential way to connect the theory to the experiments better.
> > > > > > >
> > > > > > > 3. In my opinion, the paper’s biggest contribution is showing that a hypernetwork conditioning architecture can help policies generalize better in multitask settings. The paper’s experiments and rebuttal results showed robustness and generalization well, but neither of these two properties were brought up in the theoretical section. (I think this goes into the discussion of lines 154-163 again, as it seems to mention dynamics generalization, but as I said earlier, I still don’t understand these lines of the paper and trust the authors will clarify it in the final version.)
> > > > > > >
> > > > > > > This leads me to conclude that the paper may be as effective at conveying the main point (hypernetworks are a robust way to do context-conditioning with good generalization properties) even if the authors drop the theoretical section.
> > > > > > >
> > > > > > > **Finally, some visual suggestions for improving the paper:**
> > > > > > >
> > > > > > > - I would appreciate a better color scheme for the rebuttal graphs; three of the lines looked really similar to me: adapter, cGate, and cGateEveryLayer. (Especially people without perfect color vision, avoiding having red/green/brown in the same graph, where the lines are close to each other, would help a lot.)
> > > > > > > - Is evaluation time-consuming? Graphs would look better if they didn’t only have 6-7 points per line total.
> > > > > > > - Polish up figure 1 to make it more aesthetic.

---

### Official Review · Reviewer_8dZQ · 2023-07-06

**Soundness:** 3 good
**Presentation:** 3 good
**Contribution:** 3 good
**Rating:** 7
**Confidence:** 3

**Summary:**

The paper presents an architecture that uses a hypernetwork to learn network parameters for a context-dependent policy adapter. The authors discuss an illustrative example that emphasizes in what cases a universal (context-dependent) policy is necessary, and in which cases a non-context-conditional ("unaware") policy is sufficient. In the experiments, they compare their adapter approach with a context-unaware policy, a policy that concatenates context information with state and processes them equally, and other approaches using a linear policy adapter (FLAP and cGate).

**Strengths:**

The idea to hyper-learn a nonlinear context-dependent adapter as the final few layers of a context-general policy is original and makes sense. The paper is well written, and experiments are quite thorough. The relatively detailed ablation experiments, some of which are found in the appendix, are quite interesting and a strength of the paper, for example the study of nuisance context parameters in section 7.2. I also think that the distinction between extrapolating and interpolating generalization is important and rarely seen (presumably because this distinction is usually hard to make in more complex environments), and interesting enough to maybe move it to the main part. General policies are an important and active research field, and the paper makes a significant contribution.

**Weaknesses:**

I appreciate the illustrative example for unaware vs aware policies in section 4, but I feel like this is quite intuitive for many readers, and could potentially be shortened a bit in favor of more experimental results in section 5. Also, I was a little confused by the section title being "Theoretical Foundations".

I think that the related work section should probably discuss recent advances in foundation-model-style generalist agents (e.g. https://arxiv.org/abs/2203.11931, https://arxiv.org/abs/2305.10912, https://arxiv.org/abs/2301.09816, https://arxiv.org/pdf/2306.11706.pdf) as well. The authors briefly touch on "learning some unsupervised representation of the problem and inferring a latent context from a sequence of environment observations", which is similar in spirit to this line of work, but then only cite one reference. Considering that this field has been quite active lately, it probably would make sense to discuss this briefly.

Generally, I think the paper would benefit from experiments in a larger and more diverse number of experiments. However, I think this is at least partly compensated by the fact that the authors report relatively detailed results for the two (or three) environments they consider.

Minor comments:
- The shaded areas in Fig 3 and elsewhere are very hard to read

**Questions:**

- In 6.1: Would it not be possible to calculate the average evaluation reward only on an entirely unseen set?

**Limitations:**

The authors discuss limitations in quite some detail in section 8, and underpin this with additional experiments in the appendix. This is a strength of the paper.

---

> ### Author Rebuttal · Authors · 2023-08-09
>
>
> We thank the reviewer for their kind words. The endorsement of our work is greatly appreciated. We hope to address the reviewer's remaining concerns and increase the certainty of the reviewer in their endorsement.
>
> On the point of shortening section 4: Thank you for noting the benefit of this section. We do believe section 4 motivates the need for our work - by illustrating a case where the unaware model fails. However, section 4 also notes that always being context aware is not entirely necessary. This is a necessary point in a work which aims to answer *when and how* to incorporate context. Moreover, we present a strong theoretical result which shows that, in a case where context is necessary, ignoring it becomes more costly as the number of different contexts increases. However, we will shorten this section while still keeping the illustrative and rigorous essence of what we have now.
>
> On the point of the foundation-model-style generalist agents: We thank the reviewer for directing our attention to this important area of work. We will certainly add this to our related work and believe this grounds our work further in the literature.
>
> On the point of having more experimental domains: We believe the three domains we considered represent a broad range of environments and constitute a representative selection of environments. In particular, the ODE domain is a particularly general environment, as many problems can be phrased as ODEs. Furthermore, CartPole is a relatively simple classic control environment that is widely used in the literature [1,2]. Finally, the Ant domain is a more complex and high-dimensional continuous control task and makes the applicability of our work to robotics explicit. We appreciate that the reviewer does acknowledge the detailed nature of our results in these domains, and also for their time in reading the Appendix to note the depth of our results. Indeed, there is a trade-off between the breadth of experimentation and depth of results, given a conference length page limit. We hope that we have navigated this trade-off appropriately by deeply exploring the representative and general domains. We will aim to bring more of our additional experiments into the final version of our paper - as the reviewer recommends - which we hope will further justify our experimental strategy. We thank the reviewer for their guidance on this point.
>
> On the point of the shading in, among others, Fig. 3: We will rectify this in the revised paper. We thank the reviewer for pointing this out.
>
> Finally, on the reviewer's question regarding the definition of the AER: It would be possible, however the results do not differ significantly (see Figure 4 in the attached PDF). We opted for averaging over the full context range as it provides a better metric of the general performance of the agent, however we can include the figures for the revised version.
>
>
> *References*
>
> [1] Steindór Sæmundsson, Katja Hofmann, and Marc Peter Deisenroth. Meta reinforcement learning with latent variable gaussian processes. In Amir Globerson and Ricardo Silva, editors, Proceedings of the Thirty-Fourth Conference on Uncertainty in Artificial Intelligence, pages 642–652. AUAI Press, 2018.
>
> [2] Carolin Benjamins, Theresa Eimer, Frederik Schubert, Aditya Mohan, André Biedenkapp, Bodo Rosenhahn, Frank Hutter, and Marius Lindauer. Contextualize me - the case for context in reinforcement learning. CoRR, abs/2202.04500, 2022

---

> > ### Comment · Reviewer_8dZQ · 2023-08-21
> > **Thank you, I maintain my positive score**
> >
> > I thank the authors for their detailed response. My initial concerns were only minor, and I thank the authors for adding the improvements they suggested. I maintain my positive score.

---

> > > ### Author Response · Authors · 2023-08-21
> > > **Thank you**
> > >
> > > We thank the reviewer once again for their consideration, assistance in improving the paper and overall endorsement of our work.

---

### Author Rebuttal · Authors · 2023-08-09

We thank all of the reviewers for their time and effort in reviewing our paper. We appreciate the reviewers' thoughtful comments, suggestions for improvement and generally kind words. In particular, we appreciate several of the reviewers praising the depth of our limitations section. We believe this is an important part of our work and are pleased that this was not overlooked.

There were some points raised by multiple reviewers, which we have addressed in the individual responses, but we will also add a global response here.

**Section 4/Theorem 4.5**
We originally added section 4 to motivate the need for context-aware policies, by illustrating a case (i.e., some conditions) where the unaware model fails. Section 4 also notes that always being context aware is not entirely necessary. We believe this is a necessary point in a work which aims to answer *when and how* to incorporate context. Furthermore, we added the explanation in lines 154-163 for completeness, to show how the theoretical problem we consider in section 4 is applicable to the problem of generalising to new dynamics that we consider in the rest of the paper. However, we realise that, due to the intuitive nature of our theoretical setup, this is already intuitive to most readers. Thus, we will move this to the appendix (for readers who still seek such rigour) but agree that the space could be used for other purposes.

However, based on all of the reviewers' comments, we will shorten this section (among others, moving lines 154-163 to the appendix, etc.) to make room for additional empirical results, without losing the essence of what we have now. Furthermore, we believe this section can be better contextualised and linked to our results. For instance, Figure 3 - where the Unaware model fails - on the ODE domain is an empirical illustration of Theorem 4.5(i). Figure 7 in the appendix illustrates that CartPole is an example of Theorem 4.5(ii), as the unaware model can learn a general policy that works on several contexts.

**AER Metric using only the testing contexts**
Regarding the primary metric we use to measure performance, the *Average Evaluation Reward (AER)*, we originally defined it over the entire testing context range, of which the training contexts were a subset of. We can readily change the definition of this to exclude the training contexts. However, since there are often only a handful of training contexts, the results would not look very different visually. Moreover, the conclusions will be identical. To demonstrate this fact we have added a new figure to the attached PDF containing the same result as Fig 3. in the main paper, just without the training contexts (Figure 4). We can also replace the figures in the current draft of our paper with these new ones, if so desired.


**New Rebuttal Figures**
In the attached one-page PDF, we have four new figures. We make it clear which reviewer requested each figure, however, we hope that they are of interest to all reviewers and further strengthen our submission. We would like to once again thank the reviewers for their thoughtful comments as we intend to include many of these new results in our revised draft. The figures are as follows:

1. (Figure 1) [Reviewer sPQr] Results on the 1D and 2D ODE domains of the suggested ablation methods.
2. (Figure 2) [Reviewer DXBg] The results of cGate on CartPole and Ant when adding distractor variables.
3. (Figure 3) [Reviewer sPQr] One of the adapter ablation plots that we will add to the revised version (in addition to several others, we show just this one for space reasons). This figure compares the effect of changing the Hypernetwork's architecture.
4. (Figure 4) [Reviewer 8dZQ, rS79] Showing the result of the AER metric using only the testing contexts. These results are very similar to the original results. Most importantly, our original conclusions all remain unchanged.

---

### Comment · Area_Chair_USKw · 2023-08-18
**Completing discussion on this paper**

Dear Reviewers,

Thank you all for your reviews and also for those of you who have engaged in the discussion already. If you haven't already, please take a few minutes to read the author responses and let them know if your concerns are addressed or you would like further discussion. All discussion has to be wrapped up by Monday.

Thanks,
Your AC

---

### Decision · Program_Chairs · 2023-09-21

**Decision:**

Accept (poster)

**Comment:**

This paper studies the problem of learning policies that are performant under many different system dynamics (e.g., walking on different surfaces). The paper theoretically analyzes when being context-aware is useful and when it is unecessary. This analysis is done for the limited case of goal-conditioned learning. The paper then introduces an adapter module on a policy network with weights that are set based upon the context. An empirical study shows that this architecture for context-aware policy learning is effective at training generalizable policies.

Main Strengths: Reviewers found the approach novel and the use of hyper-networks within RL to be a significant contribution.

Main Concerns: The main weakness I note is that the theoretical analysis seems limited as it only focuses on the goal-conditioned setting whereas the paper is really focusing on more general contexts. One reviewer (DXBg) noted some similarities with an earlier work and recommends that these are discussed. The similarities did not limit novelty sufficiently for this to be a major concern.

Overall, the paper introduces a novel neural network architecture that may be of broader interest in the RL community and the paper shows this architecture to be useful for training more generalist policies. I recommend acceptance.